# VE-cadherin junction dynamics in initial lymphatic vessels promotes lymph node metastasis

Miguel Sáinz-Jaspeado[1], Sarah Ring[1], Steven T Proulx[2,3], Mark Richards[1], Pernilla Martinsson[1], Xiujuan Li[4], Lena Claesson-Welsh[1], Maria H Ulvmar[1,5], Yi Jin[1]

The endothelial junction component vascular endothelial (VE)–cadherin governs junctional dynamics in the blood and lymphatic vasculature. Here, we explored how lymphatic junction stability is modulated by elevated VEGFA signaling to facilitate metastasis to sentinel lymph nodes. Zippering of VE-cadherin junctions was established in dermal initial lymphatic vessels after VEGFA injection and in tumor-proximal lymphatics in mice. Shape analysis of pan-cellular VE-cadherin fragments revealed that junctional zippering was accompanied by accumulation of small round-shaped VE-cadherin fragments in the lymphatic endothelium. In mice expressing a mutant VEGFR2 lacking the Y949 phosphosite (*Vegfr2*[Y949F/Y949F]) required for activation of Src family kinases, zippering of lymphatic junctions persisted, whereas accumulation of small VE-cadherin fragments was suppressed. Moreover, tumor cell entry into initial lymphatic vessels and subsequent metastatic spread to lymph nodes was reduced in mutant mice compared with WT, after challenge with B16F10 melanoma or EO771 breast cancer. We conclude that VEGFA mediates zippering of VE-cadherin junctions in initial lymphatics. Zippering is accompanied by increased VE-cadherin fragmentation through VEGFA-induced Src kinase activation, correlating with tumor dissemination to sentinel lymph nodes.

## Introduction

Interstitial fluid is collected by blind-ended initial lymphatic vessels and transported further to precollecting and collecting lymphatic vessels back to the blood circulation (Alitalo, 2011). The uptake of fluid by initial lymphatic vessels is driven by pressure gradients and cyclic compression and expansion of the capillaries (Schmid-Schönbein, 2003). Lymph uptake is further facilitated by the organization of lymphatic endothelial junctions in alternating regions of high and low abundance of vascular endothelial (VE)–cadherin, denoted button junctions (Baluk et al, 2007; Yao et al, 2012). A range of diseases are characterized by poor drainage of lymphatic capillaries and/or leakiness of collecting vessels, leading to establishment of edema and progression of the disease.

Cancer is a prominent example of a disease accompanied by a disorganized vasculature leading to edema, inflammation, and metastatic spread (Stylianopoulos et al, 2018). Most cancer types disseminate through the lymphatic vessels to regional lymph nodes, in addition to hematogenic spread to distant locations (Leong et al, 2011). Indeed, lymph node metastasis is an important prognostic marker for tumor progression (Stacker et al, 2002). The mechanisms that determine the initiation of metastatic spread via tumor adjacent lymphatic vessels to sentinel lymph nodes are not well understood. Expression of VEGF C or VEGFA by different cell types in the tumor correlates with increased intratumoral lymphangiogenesis and lymph node metastasis (Skobe et al, 2001). Entrance of tumor cells into the lymphatics is likely to occur at initial lymphatic vessels, facilitated by tumor-induced modulation of the lymphatic endothelium (Azzali, 2007; Farnsworth et al, 2018). Whether changes in lymphatic junctional integrity can affect the propensity for lymphatic metastasis has not been studied.

VEGFC is an important regulator for the development and growth of lymphatic vessels (Alitalo, 2011; Schulte-Merker et al, 2011). VEGFC binds to the receptor tyrosine kinase VEGF receptor-3 (VEGFR3) with high affinity, but the processed form of VEGFC can also bind to the related VEGFA receptor VEGFR2 (Joukov et al, 1997). Mouse models overexpressing VEGFC show marked increase in lymphangiogenesis and metastasis to sentinel lymph nodes (Mandriota et al, 2001; Skobe et al, 2001), and inhibition of VEGFR3 by blocking antibodies inhibits lymphangiogenesis and restricts lymph node metastasis (He et al, 2005; Burton et al, 2008). VEGFA/VEGFR2 are also important in lymphangiogenesis, promoting sentinel node metastasis in different mouse models of skin cancer (Nagy et al, 2002; Björndahl et al, 2005; Hirakawa et al, 2005). However, in tumors overexpressing VEGFA, poorly functional lymphatic vessels are created with much reduced lymphatic fluid clearance (Nagy et al, 2002).

[1]Beijer and Science for Life Laboratories, Department Immunology, Genetics and Pathology, Rudbeck Laboratory, Uppsala University, Uppsala, Sweden [2]ETH Zürich, Institute of Pharmaceutical Sciences, Zürich, Switzerland [3]Theodor Kocher Institute, University of Bern, Bern, Switzerland [4]Cyrus Tang Hematology Center, Collaborative Innovation Center of Hematology, State Key Laboratory of Radiation Medicine and Protection, Soochow University, Suzhou, China [5]Department of Medical Biochemistry and Microbiology, Uppsala University, Uppsala, Sweden

Correspondence: yi.jin@igp.uu.se

By forming homophilic interactions between blood vascular endothelial cells (BECs), VE-cadherin regulates blood vascular permeability and hematogenic metastasis (Weis et al, 2004; Li et al, 2016; Bartolome et al, 2017). Venous but not arterial blood flow and stimulation with agonists such as VEGFA leads to Src family kinase (SFK)–dependent, elevated phosphorylation of VE-cadherin, accompanied by VE-cadherin internalization, and thereby, formation of transient and discrete gaps in paracellular junctions in the blood vasculature (Honkura et al, 2018; Jin et al, 2022). $Vegfr2^{Y949F/Y949F}$ mice, lacking the tyrosine phosphorylation site at Y949 by a knock-in mutation, display a tightened blood vascular barrier and suppressed hematogenic metastasis (Li et al, 2016).

Similar to blood endothelial cells, VE-cadherin plays an important role in maintaining lymphatic junctional integrity (Baluk et al, 2007; Zheng et al, 2014; Hägerling et al, 2018). The distribution of VE-cadherin varies in different regions of the lymphatic vasculature, where initial lymphatic vessels exhibit discontinuous "button" junctions, whereas collecting lymphatic vessels are governed by continuous "zipper" junctions (Baluk et al, 2007). Although initial lymphatic vessels formed during lymphatic development are lined by continuous zipper junctions, these junctions are progressively remodeled to give rise to intermediate junctions that develop further into discontinuous button junctions (Yao et al, 2012). Specific deletion of VE-cadherin from lymphatic vessels during embryogenesis results in edema and lethality (Hägerling et al, 2018). In contrast, its deletion in the adult results in very different phenotypes dependent on the organ. In the dermis, lymphatic vessel junctions remain intact, whereas in the mesentery, lymphatic vessels deteriorate (Hägerling et al, 2018).

The aim of this study was to investigate whether lymphatic adherens junctions undergo remodeling in response to VEGFA, akin to the dynamic remodeling of blood vascular adherens junctions exposed to this growth factor. We show that VE-cadherin transforms to a fragmented morphology in peritumoral lymphatic vessels, or in vessels exposed to acute VEGFA stimulation, in the WT mouse. In contrast, VE-cadherin fragmentation is suppressed in $Vegfr2^{Y949F/Y949F}$ mice in which signaling downstream of the phosphosite Y949 is eliminated. Still, zippering of lymphatic junctions was established in these different conditions, independent of Src kinase activation. This apparent stability of lymphatic junctions in the tumor-bearing $Vegfr2^{Y949F/Y949F}$ correlates with reduced tumor cell entry into lymphatics and reduced metastatic spread to sentinel lymph nodes in orthotopically implanted melanoma and breast cancer. Combined, these results indicate that signaling via VEGFA/VEGFR2 regulates not only blood vascular permeability but also junctional integrity in initial lymphatic vessels.

## Results

### The dermal lymphatic vasculature is unperturbed in $Vegfr2^{Y949F/Y949F}$ mice

The blood vasculature in the $Vegfr2^{Y949F/Y949F}$ C57Bl/6 model develops normally, but the BECs are resistant to the permeability enhancing effect of VEGFA (Li et al, 2016). As lymphatic endothelial cells (LECs) express considerable levels of VEGFR2 (Wirzenius et al,

2007), it is important to understand the consequence of loss of the VEGFR2 phosphosite Y949 and its downstream signaling on lymphatic vessel development and lymphatic function.

To verify that lymphatic vessels in the $Vegfr2^{Y949F/Y949F}$ mouse develop normally, embryos at embryonic (E) day 14.5 were examined. Neuropilin2 (Nrp2)$^+$ lymphatic tip cell numbers and lymphatic vessel density in the dorsal skin were similar in $Vegfr2^{Y949F/Y949F}$ and WT littermates (Fig 1A–C). Moreover, analyses in 8–10-wk-old mice showed similar lymphatic vessel endothelial hyaluronan receptor 1 (LYVE1)$^+$ vessel diameter in the ear dermis of $Vegfr2^{Y949F/Y949F}$ and WT mice, whereas vessel density was increased in the $Vegfr2^{Y949F/Y949F}$ genotype (Fig 1D–F). Expression levels of VEGFR2, VEGFR3, and Podoplanin in LECs isolated from postnatal (P) day 10 lungs were not affected by the Y949F mutation (Fig 1G and H). We conclude that elimination of the phosphosite Y949 in VEGFR2 by replacing tyrosine (Y) with phenylalanine (F) did not perturb lymphatic development or the expression of VEGF receptors in LECs.

### Tumor growth and lymphatic metastasis in $Vegfr2^{Y949F/Y949F}$ mice

The VEGFA-resistant vascular barrier in the $Vegfr2^{Y949F/Y949F}$ mouse blood vasculature correlates with decreased hematological metastatic spread from RipTag2 neuroendocrine tumors or B16F10 melanoma (Li et al, 2016). To explore the effect of the $Vegfr2^{Y949F/Y949F}$ mutation on lymphatic vessel barrier function in cancer, DsRed-expressing B16F10 melanoma cells were engrafted intradermally in the ear and analyzed after 7 or 12 d of tumor growth. The ear dermis was chosen as the site of injection as it restricts local growth and promotes spread to sentinel lymph nodes (Li et al, 2016). A low inoculation volume (5 $\mu l$) was used to minimize tissue damage and forced metastasis. There was no difference in the growth rate or weight at day 12 of the primary B16F10 tumors between the $Vegfr2^{Y949F/Y949F}$ and WT mice (Fig 2A and B). However, the weight of sentinel cervical lymph nodes was significantly lower in the $Vegfr2^{Y949F/Y949F}$ mouse (Fig 2C). In accordance, expression of $Tyrp1$ (melanocyte-specific gene) was decreased in sentinel lymph nodes of the $Vegfr2^{Y949F/Y949F}$ mice compared with WT, indicating reduced lymphatic tumor dissemination (Fig 2D).

A similar pattern was observed when challenging mice with EO771-CCR7-tdTomato mammary carcinoma, engrafted in the mammary fat pad of female mice. In this model, C-C chemokine receptor type 7 (CCR7) expression was introduced in the EO771 tumor cell line to promote lymph node metastasis, previously also shown to promote lymph node metastasis in other models of breast cancer (Cunningham et al, 2010). Primary tumor weight at day 20 was similar between $Vegfr2^{Y949F/Y949F}$ and WT mice (Fig 2E). Although the weight of sentinel inguinal lymph nodes was unaffected (Fig 2F), flow cytometry analysis of tumor cells positive for tdTomato in the inguinal lymph nodes showed significantly fewer tumor cell counts in the $Vegfr2^{Y949F/Y949F}$ compared with the WT lymph nodes (Fig 2G and H). Of note, immune cells, potentially positive for tdTomato by uptake of tumor debris, were excluded in the analysis.

Lymphangiogenesis occurs preferentially at the tumor periphery in response to tumor-secreted growth factors (Christiansen & Detmar, 2011). The Y949F mutation did not affect tumor-induced lymphangiogenesis, as shown by the unperturbed LYVE-1+ lymphatic vessel density at the tumor periphery in the B16F10-

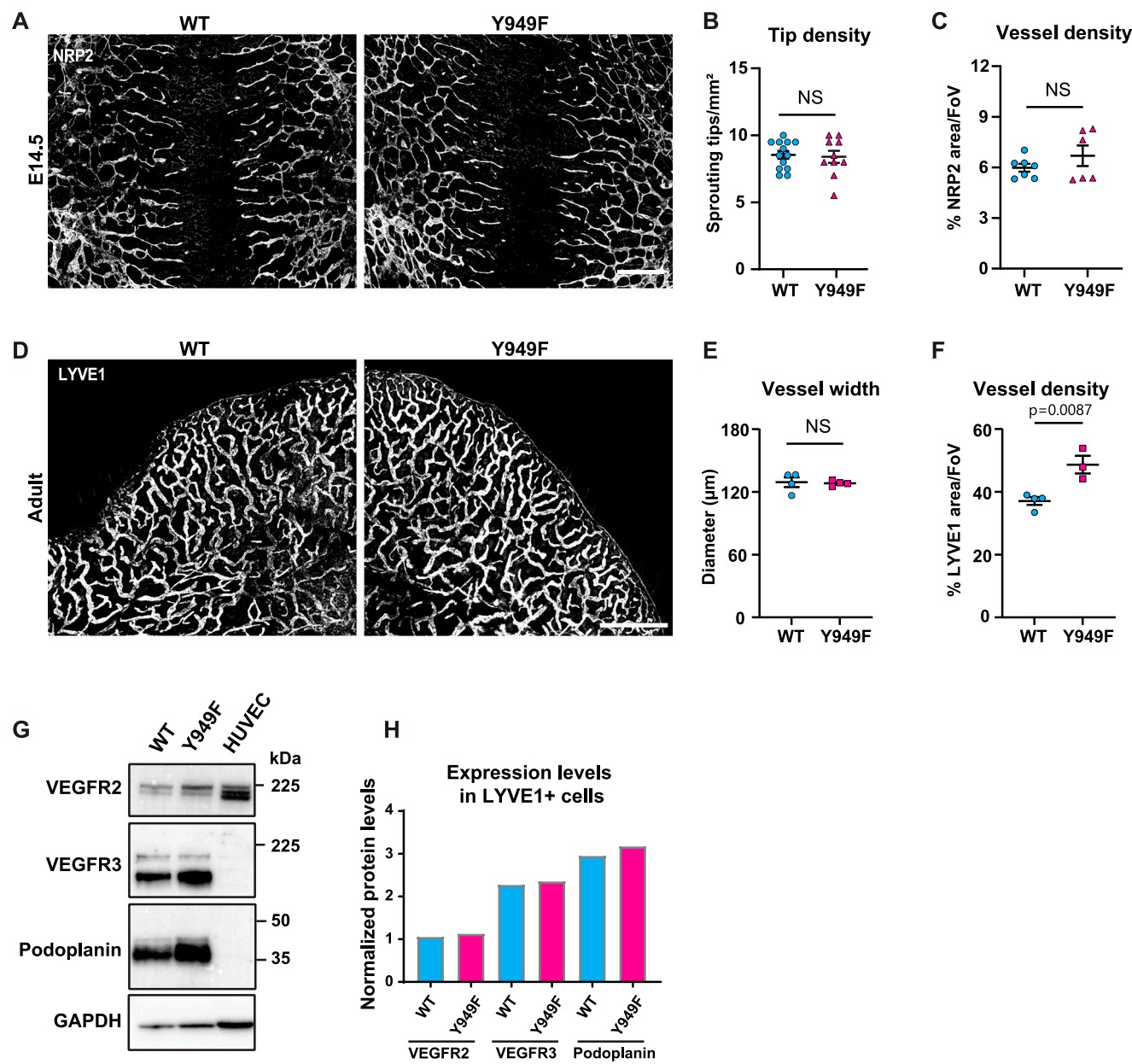

**Figure 1. Characterization of dermal lymphatic vessels in WT and _Vegfr2^Y949F/Y949F^_ embryos and adult mice.**
**(A)** Dermal lymphatic vessels in E14.5 WT and _Vegfr2^Y949F/Y949F^_ (Y949F) embryos shown by immunostaining of Neuropilin2 (Nrp2) in the back skin. Scale bar, 500 μm. **(B)** Quantification of dermal lymphatic sprouting tips/mm² in the E14.5 embryos. WT, n = 13; Y949F, n = 10. NS, not significant, _t_ test. **(C)** Lymphatic vessel density measured by NRP2 staining and normalized to tissue area/field of view in the E14.5 embryos. WT, n = 7; Y949F, n = 6. NS, not significant, _t_ test. **(D)** Representative images of LYVE1 immunofluorescence showing ear dermal lymphatic vasculature in 8–10-wk-old mice. Scale bar, 1 mm. **(E)** Quantification of LYVE1⁺ lymphatic vessel width in 8–10-wk-old mouse ear dermis. n = 4 mice/genotype. NS, not significant, _t_ test. **(F)** Quantification of LYVE1⁺ lymphatic vessel density in the 8–10-wk-old mouse ear dermis. WT, n = 4; Y949F, n = 3. P = 0.0087, _t_ test. **(G)** Expressions of VEGFR2, VEGFR3, and Podoplanin in isolated LYVE1⁺ cells from lungs of WT and _Vegfr2^Y949F/Y949F^_ mice at postnatal day 10 and in HUVECs, detected by immunoblotting. **(H)** Quantification of expression levels normalized to GAPDH in (G).
Source data are available for this figure.

engrafted _Vegfr2^Y949F/Y949F^_ and WT mouse ears (Fig S1A and B). Moreover, there was a trend but no significant difference in expressions of _Vegfa_ and _Cdh5_ in _Vegfr2^Y949F/Y949F^_ and WT B16F10 tumors (Fig S1C and D). We conclude that in two different orthotopic cancer models, breast cancer and melanoma, metastatic spread to sentinel lymph nodes was suppressed in mice lacking the VEGFR2 Y949 phosphosite.

## Lymphatic drainage and intravasation of tumor cells in _Vegfr2^Y949F/Y949F^_ mice

The potential consequence of the Y949F mutation on lymphatic function was addressed by monitoring the clearance of the near-infrared tracer P20D800 by intravital imaging (Proulx et al, 2013) of the mouse ear. The movement of interstitial fluid into

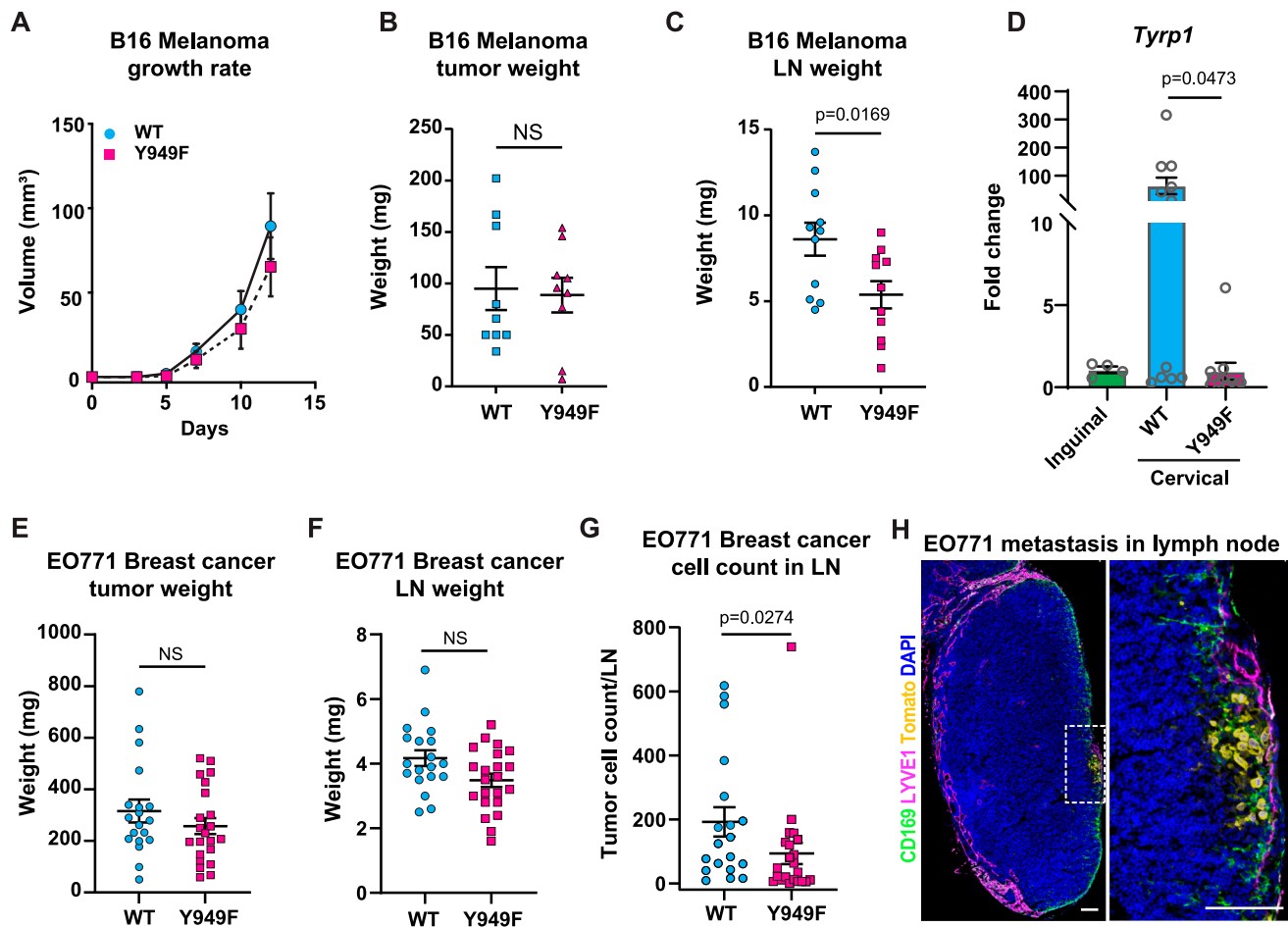

**Figure 2. B16F10 melanoma and EO771-CCR7 tumor growth and lymph node metastasis.**
**(A, B)** B16F10 melanoma growth rate (A) and weight at day 12 after inoculation (B) of adult WT and *Vegfr2$^{Y949F/Y949F}$* (Y949F) in the ear dermis. Repeated measurements of tumor volume were evaluated using ANOVA with regard to genotype and time, which showed no significant (NS) difference between WT and Y949F mice. n = 9 mice/genotype. **(C)** Sentinel cervical lymph node weights at day 12 after inoculation, harvested from mice with B16F10 tumors. n = 11 lymph nodes/genotype, $P = 0.0169$, t test. LN, lymph node. **(D)** Detection of *Tyrp1* gene expression by quantitative PCR in cervical lymph nodes at day 12 of tumor growth normalized to unaffected inguinal lymph nodes. Ribosomal protein gene *Rpl19* was used as the internal control. WT, n = 9 lymph nodes; Y949F, n = 10 lymph nodes, $P = 0.0473$, t test. **(E)** Primary tumor weight at day 20 after engraftment of EO771-CCR7-tdTomato mammary cancer cells. WT, n = 19 mice; Y949F, n = 22 mice. NS, not significant, t test. **(F)** Weights of inguinal lymph node (LN) isolated at day 20 from WT and Y949F mice transplanted with EO771-CCR7-tdTomato mammary cancer cells. WT, n = 19 mice; Y949F, n = 22 mice. NS, not significant, t test. **(G)** Count of td-Tomato–positive EO771 cells in inguinal lymph nodes (LN) by flow cytometry. Mann–Whitney test, $P = 0.0274$. WT, n = 19 mice; Y949F, n = 22 mice. **(H)** Representative image of lymph node from WT mouse showing metastasis of EO771-CCR7-tdTomato mammary cancer cells. Tumor cells (tdTomato+) are highlighted in boxed region and shown enlarged on the right, combined with CD169 (macrophage, green) and LYVE1 (lymphatic endothelium, magenta). Scale bars, 100 *μ*m. Source data are available for this figure.

the lymphatic vessels (lymphatic clearance) is mediated by both interstitial fluid pressure and the pumping created by the smooth muscle cells surrounding the collecting vessels (Bazigou et al, 2014). Lymphatic clearance and half-life of the tracer was not significantly different between healthy mutant and WT mice (Fig 3A and B). However, after engraftment of B16F10 melanoma, lymphatic clearance of the fluorescent tracer was faster in *Vegfr2$^{Y949F/Y949F}$* mutant mice with reduced half-life compared with WT mice (Fig 3A and B).

Movement of tumor cells into initial lymphatic vessels is dependent on interstitial pressure gradients but also on chemokines produced in the microenvironment and the expression of adhesion molecules on LECs (Shields et al, 2007; Swartz & Lund, 2012). The

presence of metastatic melanoma cells within the peritumoral lymphatic vessels was determined by counting the number of DsRed-expressing B16F10 cells, using orthogonal projection of confocal imaging. Tumor cells were more frequently found inside lymphatic capillaries of WT mice at day 7 than in *Vegfr2$^{Y949F/Y949F}$* mice (Fig 3C and D).

The similar lymphatic clearance rate by unchallenged WT and *Vegfr2$^{Y949F/Y949F}$* lymphatic vessels suggests no functional deviation because of the VEGFR2 mutation. When subjected to tumor challenge, the *Vegfr2$^{Y949F/Y949F}$* mutation led to improved lymphatic drainage and reduced transmigration of tumor cells into peritumoral lymphatic vessels, indicating an enhanced lymphatic barrier.

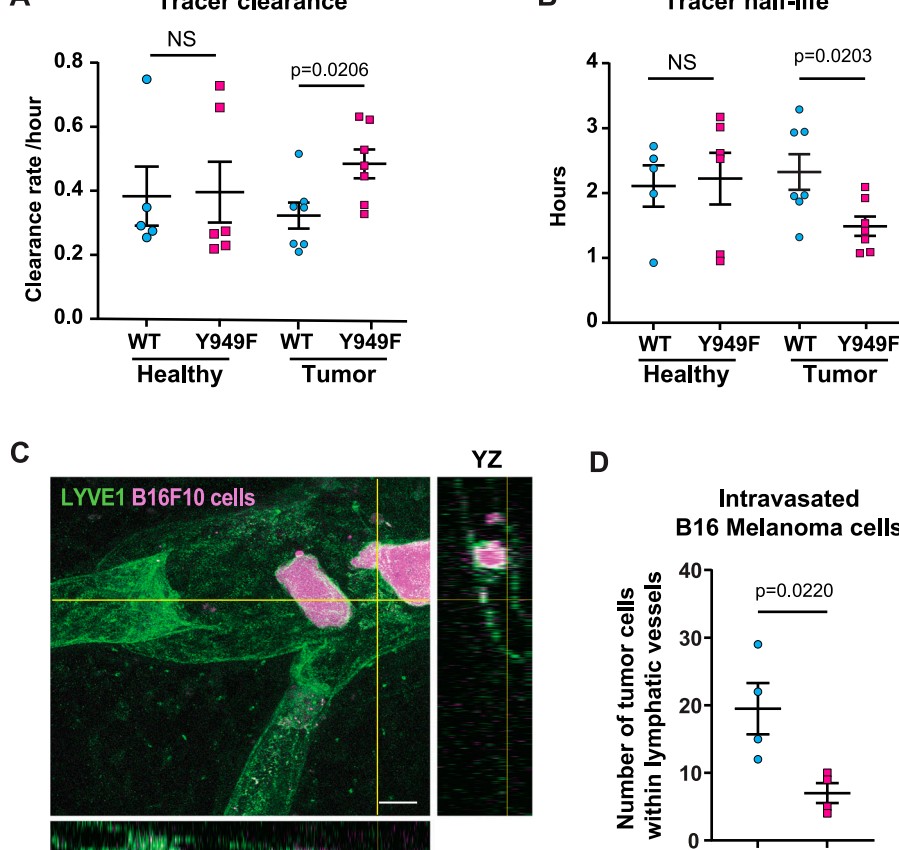

**Figure 3. Lymphatic drainage and tumor cell intravasation in lymphatic vessels.**
**(A, B)** Lymphatic clearance rate (A) and half-life (B) of the near-infra red dye P20D800 in the ear dermis of healthy and B16F10 melanoma tumor–engrafted mice. WT healthy mice, n = 5; Y949F healthy mice, n = 6; WT tumor mice, n = 7; Y949F tumor mice, n = 7. The t test was used for statistical evaluation. NS, not significant. **(C)** Representative image showing the lymphatic vessel in the WT ear dermis immunostained for LYVE1 (green) with intravasated B16F10<sup>DsRed+</sup> melanoma cells (magenta) inside or entering the vessel. Orthogonal views (YZ and XZ) shown to the right and below visualize cells within the vessel. Scale bar, 20 μm. **(D)** Quantification of B16F10<sup>DsRed+</sup> melanoma cells present inside peritumoral lymphatic vessels at day 7 after tumor cell engraftment. P = 0.022, n = 4 mice/genotype, t test. Source data are available for this figure.

## Characterization of lymphatic endothelial junctions in *Vegfr2^{Y949F/Y949F}* mice

We hypothesized that the improved interstitial fluid clearance in the melanoma-bearing *Vegfr2^{Y949F/Y949F}* ear and the reduced metastatic spread observed in this model may be related to stabilization of VE-cadherin in the lymphatic endothelial junctions. VE-cadherin junctions undergo "zippering" in response to VEGFA (Zhang et al, 2018; Zarkada et al, 2023), that is, conversion to a linear and continuous VE-cadherin immunostaining pattern. By analyzing the pattern of VE-cadherin–positive LEC junctions, it was evident that zippering of the lymphatic junctions was established at the tumor periphery in both WT and *Vegfr2^{Y949F/Y949F}* mutant mice (Fig 4A and B). The extent of zippering was slightly lower in the mutant mice, as shown by the extent of VE-cadherin coverage at the cell perimeter (Fig 4B). Increased lymphatic zippering in the tumor-proximal lymphatic vessels compared with vessels distal from the tumor was also evident from the decrease in the number of VE-cadherin fragments and increase in the fragment length (Fig 4C and D).

In BECs, phosphorylation of VE-cadherin at Y685 marks vessels susceptible to increased permeability when appropriately stimulated with agonists such as VEGFA (Claesson-Welsh et al, 2021). Immunostaining for pY685 VE-cadherin using a validated antibody

(Jin et al, 2022) showed robust reactivity in dermal lymphatic vessels; however, there was no difference in signal strength between the WT and *Vegfr2^{Y949F/Y949F}* ear dermis (Fig S2A and B).

Uptake of interstitial fluid correlates with the dynamic turnover and distribution of VE-cadherin (Trzewik et al, 2001; Schmid-Schönbein, 2003; Baluk et al, 2007), which is reflected in the morphology of VE-cadherin as visualized by immunofluorescent staining. To follow VE-cadherin dynamics in a more sensitive, unbiased manner throughout the cell, we developed an automated classification of the shapes of VE-cadherin fragments based on the aspect ratio and circularity. Four categories of VE-cadherin shapes were defined: category 1—fragments with an aspect ratio above the upper quartile (Q3) and circularity between 0–0.25; category 2—aspect ratio Q2–Q3 and circularity 0.25–0.5; category 3—aspect ratio Q1–Q2 and circularity 0.5–0.75; category 4—aspect ratio below Q1 and circularity 0.75–1 (Fig 4E). The short round shapes in categories 3–4 may result from dynamic turnover of VE-cadherin, that is, internalization (Bentley et al, 2014).

Applying this classifier to the healthy ear dermal LYVE1<sup>+</sup>/VE-cadherin<sup>+</sup> lymphatic vessels showed that there was no difference in the frequency of VE-cadherin<sup>+</sup> categories when comparing un-challenged WT and *Vegfr2^{Y949F/Y949F}* dermis (Fig 4F). However, the frequency of category 4 VE-cadherin shapes increased, whereas category 3 shapes decreased, in the melanoma-proximal lymphatic

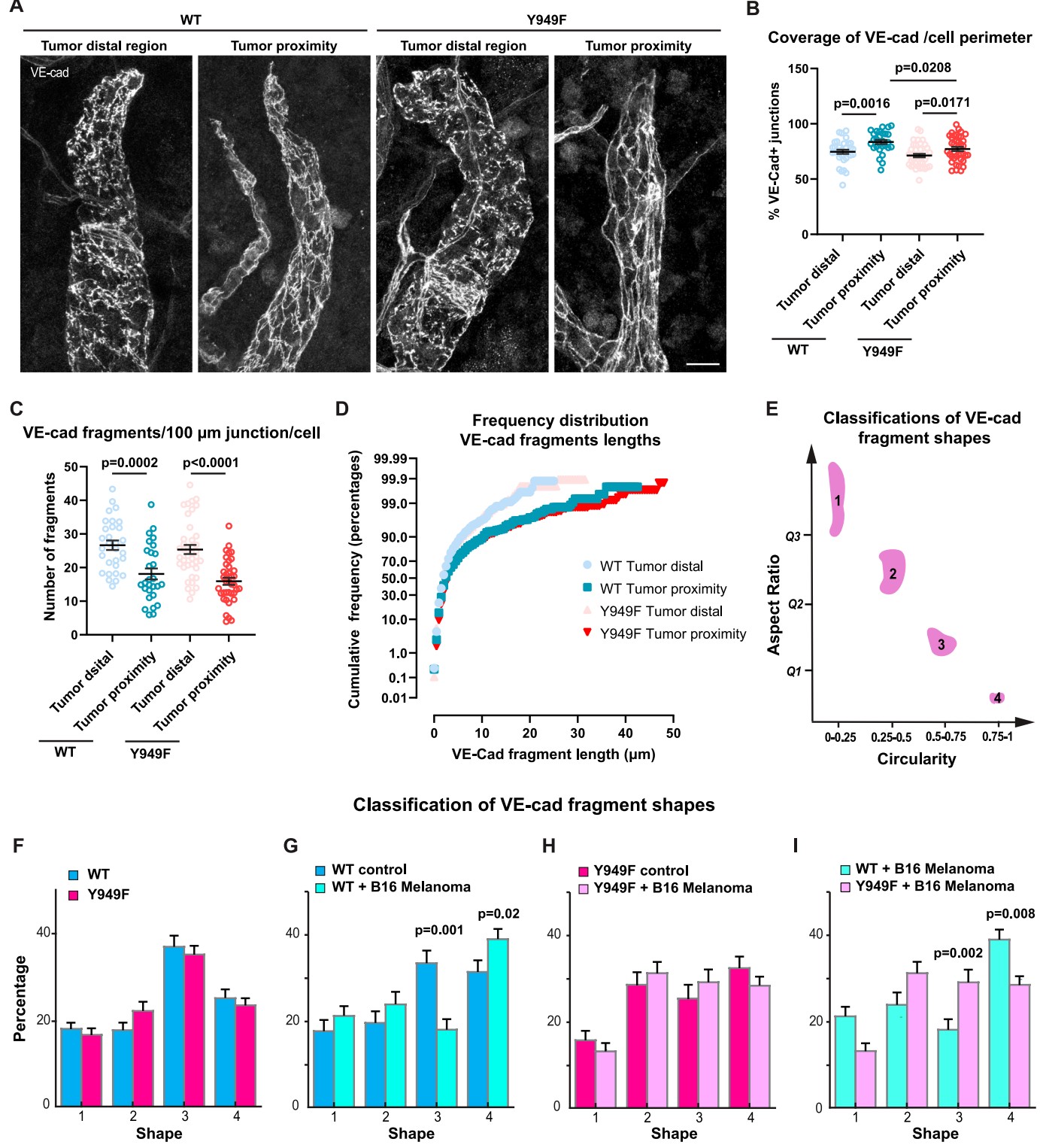

**Figure 4. Characterization of lymphatic VE-cadherin junctions in tumor-engrafted ear dermis.**
**(A)** Junctions in initial lymphatic vessels in tumor distal and tumor proximal regions visualized by immunostaining of VE-cadherin. Samples are from WT or *Vegfr2^{Y949F/Y949F}* ears, either healthy (control) or from the peritumoral region of B16F10 tumors at day 7 after inoculation. Scale bar, 20 μm. **(B)** Quantification of VE-cadherin coverage at the perimeter of LECs in initial lymphatic vessels at tumor-proximal and distal regions. Data show percentage of VE-cadherin coverage at the junction of each cell (summed junctional VE-cadherin fragment length/cell perimeter × 100%). WT tumor distal, n = 33 cells analyzed from 4 mice, WT tumor proximal, n = 28 cells from 4 mice; Y949F tumor distal, n = 40 cells from 4 mice; Y949F tumor proximal, n = 39 cells from 4 mice. The *t* test was used for statistical analysis. **(C)** Quantification of the number of VE-cadherin fragments at the junction of each LEC per 100 μm initial lymphatic vessel at tumor distal and proximal regions. WT tumor distal, n = 33 cells analyzed from 4 mice, WT tumor

vessels compared with healthy tissue in the WT mouse ear (Fig 4G). The more elongated VE-cadherin shapes (categoryies 1 and 2) remained unaffected. In contrast, the VE-cadherin shapes were similarly distributed irrespective of the presence of a tumor in the *Vegfr2^Y949F/Y949F^* mouse (Fig 4H). Consequently, comparing VE-cadherin fragment distribution between the genotypes showed that melanoma-proximal WT lymphatic vessels had an increased proportion of category 4 VE-cadherin shapes compared with *Vegfr2^Y949F/Y949F^* (Fig 4I).

Together, our data identified zippering of button junctions in the tumor-proximal lymphatics. Junctional zippering was accompanied by an increase in pan-cellular VE-cadherin fragments with high circularity and a low aspect ratio in the WT mouse lymphatic endothelium but not in mice expressing the Y949F mutant VEGFR2. These data indicate increased VE-cadherin dynamics in the tumor-challenged WT but not in the *Vegfr2^Y949F/Y949F^* initial lymphatics.

### Effect of VEGFA and VEGFC on dermal lymphatic vessels

Hypoxia-induced increase in VEGFA levels in the tumor microenvironment correlates with disintegration of VE-cadherin junctions and dismantling of the blood vascular barrier (Weis et al, 2004; Li et al, 2016). In lacteal and dermal lymphatic vessels, VEGFA has been shown to induce zippering of junctions through activations of VEGFR2 (Zhang et al, 2018; Zarkada et al, 2023). We hypothesized that the zippering of dermal lymphatic junctions and the shape shift of VE-cadherin fragments in the tumor periphery may be related to production of VEGFA and/or VEGFC, which both bind to VEGFR2 (Simons et al, 2016), in the tumor microenvironment. VEGFA secreted by malignant B16F10 melanoma cells vary from picogram to nanogram levels dependent on culture conditions (Collet et al, 2014; Jean et al, 2014; Palazon et al, 2017). Furthermore, serum VEGFA levels in melanoma patients vary from 100–500 pg/ml, values dependent on the disease stage (Ugurel et al, 2001; Osella-Abate et al, 2002; Pelletier et al, 2005; Palmer et al, 2011). Injection of 0.25 ng of VEGFA in 5 µl in the WT ear dermis led to zippering of lymphatic junctions with a reduced number of VE-cadherin fragments and increased fragment length (Fig 5A–D). In *Vegfr2^Y949F/Y949F^* mice, lymphatic junctions were more zippered than in the WT in basal conditions (Fig 5B), zippering increased further in both genotypes by VEGFA injection (Fig 5A and B), accompanied by a decrease in the number of VE-cadherin fragments/vessel length (Fig 5C and D).

Shape classification of pan-endothelial VE-cadherin fragments was performed to analyze the effect of VEGFA or VEGFC on the dynamics of lymphatic junctions in the healthy tissue. Injection of VEGFA (0.25 ng in 5 µl) in the WT ear dermis led to an increased

detection of category 4 VE-cadherin shapes and concomitantly, reduced category 3 shapes at 15 min postinjection, whereas elongated fragments (categories 1 and 2) remained unaffected (Fig 5E). The distribution of VE-cadherin shape categories was restored to basal 30 min after injection of VEGFA (Fig 5F), indicating transient dynamics. In accordance with the observation in the tumor-bearing mice (Fig 4H), VE-cadherin shapes were not altered in response to VEGFA injection in the *Vegfr2^Y949F/Y949F^* dermis (Fig 5G and H). Injection of VEGFC (25 ng in 5 µl) also induced increased frequency of category 4 VE-cadherin shapes in the WT but not in the *Vegfr2^Y949F/Y949F^* dermis (Fig S3A–C).

These results show that similar to the zippering of VE-cadherin junctions in tumor-proximal lymphatics, injection of VEGFA induced zippering of button junctions in dermal lymphatic vessels. The lymphatic vessels in *Vegfr2^Y949F/Y949F^* mice were resistant to VEGFA/VEGFC–induced increase in circular VE-cadherin shapes. Together, these data indicate that the Y949 site in VEGFR2 is dispensable for VEGFA-induced zippering of lymphatic junctions which is in agreement with a recent study (Zarkada et al, 2023). However, importantly, signaling downstream of Y949 in VEGFR2 is required for VEGFA/VEGFC–induced internalization and turnover of VE-cadherin.

### VEGFA-mediated VE-cadherin shape change is dependent on Src

Signaling pathways involved in VEGFA-mediated zippering of lymphatic junctions have been revealed in recent studies (Zhang et al, 2018; Zarkada et al, 2023). However, VEGFA-induced signaling regulating VE-cadherin dynamics in lymphatic vessels has not been identified. SFKs have been implicated in VEGFR2-induced modulation of blood vascular VE-cadherin junctions, by phosphorylating VE-cadherin, leading to disruption of VE-cadherin homophilic interactions followed by internalization. The role of c-Src in the regulation of lymphatic junctions was tested in vivo using a *c-Src^fl/fl^*; *Cdh5CreERT2* mouse model (*Src* iECKO) (Schimmel et al, 2020; Jin et al, 2022), in which tamoxifen treatment results in an endothelial-specific deletion of c-Src. No effect of c-Src deletion on lymphatic vessel morphology was observed within the period of observation, in agreement with the c-Src–deficient blood vasculature morphology appears normal (Jin et al, 2022). The *Src* iECKO mouse failed to respond to VEGFA with an increase in category 4 VE-cadherin fragments, which was established in the Cre negative control ear dermis (Fig 6A and B). Interestingly, with VEGFA treatment, the percentage of elongated VE-cadherin fragment (category 2) was increased in the *Src* iECKO genotype, compared with the control (Fig 6C), in accordance with more stable lymphatic endothelial junctions when Src activity was suppressed. We conclude that Src

---

proximal, n = 28 cells from 4 mice; Y949F tumor distal, n = 40 cells from 4 mice; Y949F tumor proximal, n = 39 cells from 4 mice. The *t* test was used for statistical analysis. **(D)** Cumulative frequency analysis of the lengths of junctional VE-cadherin fragments in initial lymphatic vessels in tumor distal and proximal regions. WT tumor distal, n = 784 fragments analyzed from 4 mice; WT tumor proximal, n = 433 fragments from 4 mice; Y949F tumor distal, n = 928 fragments from 4 mice; Y949F tumor proximal, n = 641 fragments from 4 mice. **(E)** Classification of shapes of VE-cadherin fragments in four categories based on the aspect ratio and circularity. **(F)** Shape analysis of pan-cellular VE-cadherin fragments in the WT and Y949F dermal lymphatic vessels. Data show percentage of different shape categories. WT, n = 5 mice; Y949F, n = 7 mice. **(G)** Shape analysis of VE-cadherin fragments in the WT dermal lymphatic vessels, challenged with B16F10 melanoma or not. Control, n = 5 mice; tumor challenged, n = 7 mice. The *t* test was used for statistical analysis. *P* = 0.02 for category 4; *P* = 0.001 for category 3 comparing healthy and tumor-bearing WT mice. **(H)** Shape analysis of VE-cadherin fragments in the ear dermis lymphatic vessels of *Vegfr2^Y949F/Y949F^* (Y949F) mice challenged or not with B16F10 melanoma. Control, n = 5 mice; tumor challenged, n = 7 mice. **(I)** Comparison of VE-cadherin fragment distribution between tumor-challenged WT and Y949F mice. n = 7 mice/genotype. *P* = 0.002, for category 3, *P* = 0.008 for category 4 comparing tumor-bearing WT and Y949F mice, *t* test.
Source data are available for this figure.

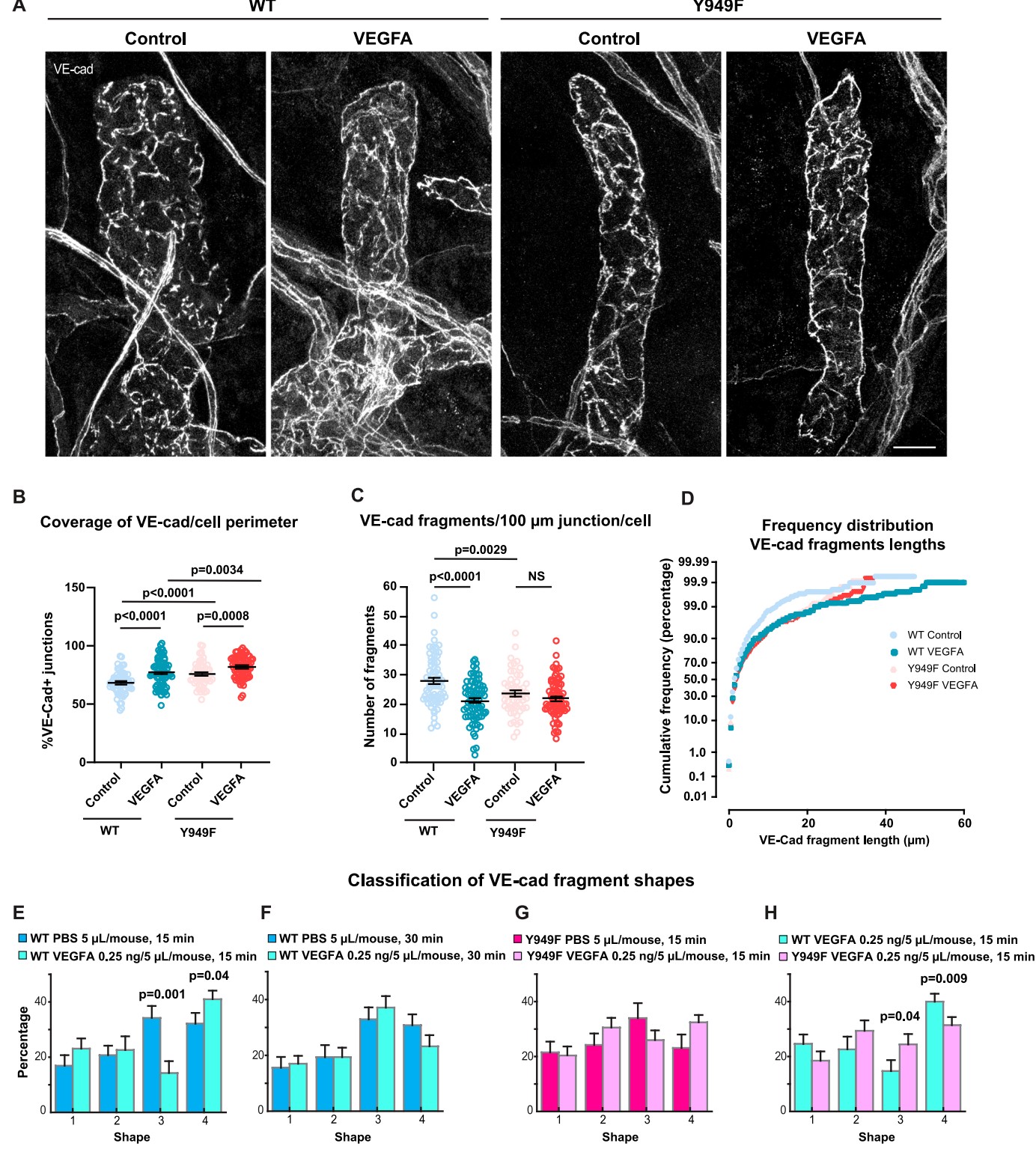

**Figure 5. VEGFA-induced VE-cadherin dynamics.**
**(A)** Junctions in initial lymphatic vessels in the ear dermis of WT and *Vegfr2*^Y949F/Y949F mice locally injected with PBS (control) or VEGFA (0.25 ng, 5 *µl*) visualized by immunostaining of VE-cadherin. Scale bar, 20 *µm*. **(B)** Quantification of VE-cadherin coverage at the perimeter of LECs in initial lymphatic vessels in the ear dermis, injected with PBS or VEGFA. Data show percentage of VE-cadherin coverage at the junction of each cell (summed junctional VE-cadherin fragment length/cell perimeter × 100%). WT control, n = 69 cells analyzed from 6 mice, WT VEGF, n = 78 cells from 6 mice; Y949F control, n = 53 cells from 5 mice; Y949F VEGF, n = 76 cells from 6 mice. The *t* test was used for statistical analysis. **(C)** Quantification of the number of VE-cadherin fragments at the junction of each endothelial cell per 100 *µm* initial lymphatic vessel in the ear dermis injected with PBS or VEGFA. WT control, n = 69 cells analyzed from 6 mice, WT VEGF, n = 76 cells from 6 mice; Y949F control, n = 53 cells from 5 mice; Y949F

expression/activity is required for VEGFA-induced increase of the circular VE-cadherin shape (category 4) in capillary lymphatic vessels.

# Discussion

Adherens junction formed by homophilic adhesion of VE-cadherin is an essential component in the lymphatic barrier. Interstitial fluid, macromolecules, and immune cells enter the lymphatic circulation through the button junctions of initial lymphatics, whereas collecting lymphatic vessels are equipped with zipper junctions that restrict exchange with the surrounding tissue (Schulte-Merker et al, 2011). Deletion of VE-cadherin in adult dermal lymphatics is accompanied by fragmentation of PECAM⁺ junctions (Hägerling et al, 2018), and perfusion of collecting vessels with a VE-cadherin blocking antibody leads to increased lymphatic permeability, allowing passage of tracers across the vessel wall (Jannaway & Scallan, 2021). Zippering of lymphatic button junctions has been shown to be induced by VEGF-mediated signaling in intestinal lacteals and in dermal lymphatics through phosphoinositide 3-kinase–mediated Rac activation, restricting cytoskeletal contraction (Zhang et al, 2018; Zarkada et al, 2023). Here, we show marked increase in VE-cadherin fragmentation indicative of increased VE-cadherin dynamics in initial lymphatic vessels in the ear dermis in the presence of a tumor and in response to VEGFA or VEGFC treatment. Concomitant with fragmentation, VE-cadherin button junctions transformed into a zipper-like morphology. Tumor or VEGF-induced zippering of lymphatic junction was not affected in a VEGFR2 mutant mouse model (Vegfr2$^{Y949F/Y949F}$) (Li et al, 2016) which lacks the Y949 residue critical for activation of SFKs in response to VEGFA. In contrast, VE-cadherin fragmentation was decreased in the Vegfr2$^{Y949F/Y949F}$ mice and in Src iECKO mice. These results are in agreement with the recent literature, showing that VEGFC treatment induces VE-cadherin phosphorylation and internalization in LECs in a c-Src–dependent manner (Sung et al, 2022). The reduced dynamics of VE-cadherin in Vegfr2$^{Y949F/Y949F}$mice correlated with decreased intravasation of tumor cells into lymphatics and establishment of lymph node metastasis in breast cancer and melanoma tumor models.

Tumor-induced zippering was comparable in the WT and the Y949F mutant strains; still, lymph node metastasis was lower in Vegfr2$^{Y949F/Y949F}$mice. This result indicates that rearrangement of lymphatic junctions to a zipper morphology does not restrict tumor cell intravasatation into lymphatic vessels. Instead, our data show that VE-cadherin fragmentation correlates with tumor cell

metastasis. To investigate junction dynamics in initial lymphatic vessels based on VE-cadherin morphologies, we applied an automated classification tool to estimate VE-cadherin fragment shapes based on their aspect ratio and circularity throughout the cell. There was no difference in the distribution of junction categories between the WT and Vegfr2$^{Y949F/Y949F}$ ear dermis in the absence of challenge such as tumor growth of VEGF stimulation (see Fig 4F). We suggest that the increased appearance of small, category 4 shapes in WT mouse LECs upon tumor challenge was at least in part because of VEGFA up-regulation in the hypoxic tumor environment and signal transduction via VEGFR2 pY949 and c-Src (Weis et al, 2004; Li et al, 2016), causing VE-cadherin internalization and degradation. Indeed, VEGFA/VEGFC–injection reproduced the changes observed in the tumor-adjacent lymphatics in the WT strain but did not cause fragmentation in the Y949F mutant lymphatics. VE-cadherin fragmentation was rapidly established (within 15 min after injection) upon injection of VEGFA/VEGFC, indicating a direct effect on lymphatic barrier properties. Although our data show that the appearance of small VE-cadherin fragments covaried with entry of tumor cells into vessels and dissemination to sentinel lymph nodes in the WT mouse, it does not provide evidences for a cause–consequence relationship between fragmentation, tumor cell entry, and metastatic spread. We speculate that the suppressed fragmentation of VE-cadherin in the Y949F mutant indicates a stabilized lymphatic barrier, hindering entry of tumor cells into lymphatics. We cannot exclude that the difference between the tumor-challenged WT and Vegfr2$^{Y949F/Y949F}$ strains in lymphatic integrity is in part influenced by the decrease in blood vessel permeability in the Vegfr2$^{Y949F/Y949F}$ strain (Li et al, 2016). Reduced blood vessel permeability would lead to slower interstitial fluid build-up, reduced interstitial pressure, and possibly also affect the entry of inflammatory cells (Claesson-Welsh et al, 2021). Production of cytokines in the tumor microenvironments promoting lymphatic metastasis such as TGF-β1 (Pang et al, 2016) could therefore be affected in Vegfr2$^{Y949F/Y949F}$mice.

Vascular permeability regulated through VE-cadherin internalization and turnover has been well described in blood vessels. VE-cadherin is constitutively internalized in blood endothelial cells through Yes kinase–mediated phosphorylation (Jin et al, 2022). Retention of VE-cadherin at the plasma membrane upon loss of Yes leads to reduced junction plasticity and increased permeability. The internalized VE-cadherin undergoes ubiquitination-dependent degradation, and deletion of ubiquitin ligase CHFR (checkpoint protein with FHA and Ring domain) blocks the degradation of VE-cadherin and enhances blood vessel integrity (Tiruppathi et al, 2023). In lymphatic vessels, regulation of permeability is conferred

---

VEGF, n = 75 cells from 6 mice. The t test was used for statistical analysis. NS, not significant. **(D)** Cumulative frequency analysis of the lengths of VE-cadherin fragments in initial lymphatic vessels after injection of PBS or VEGFA. WT PBS, n = 1,870 fragments analyzed from 6 mice, WT VEGFA, n = 1,815 fragments from 6 mice; Y949F PBS, n = 1,236 fragments analyzed from 5 mice; Y949F VEGFA, n = 1,502 fragments from 6 mice. **(E)** Shape analysis of VE-cadherin fragments in lymphatic vessels upon intradermal injection of WT ears with VEGFA (0.25 ng in 5 μl) or PBS (5 μl) followed by fixation at 15 min after injection. P = 0.04 for category 4, P = 0.001 for category 3. PBS, n = 5 mice; VEGFA, n = 5 mice; the t test was used for statistical analysis. **(E, F)** Shape analysis of VE-cadherin fragments in lymphatic vessels as in (E) but with fixation at 30 min after injection. PBS, n = 5 mice; VEGFA, n = 5 mice. **(G)** Shape analysis of VE-cadherin fragments in lymphatic vessels upon intradermal injection of Vegfr2$^{Y949F/Y949F}$ (Y949F) ears with VEGFA (0.25 ng in 5 μl) or PBS (5 μl) followed by fixation at 15 min after injection. PBS, n = 7 mice; VEGFA, n = 7 mice. The t test was used for statistical analysis. **(H)** Comparison of the effect of VEGFA administration between WT and Vegfr2$^{Y949F/Y949F}$ genotypes. WT VEGFA, n = 5 mice; Y949F VEGFA, n = 7 mice. P = 0.04 for category 3, P = 0.009 for category 4, t test. Source data are available for this figure.

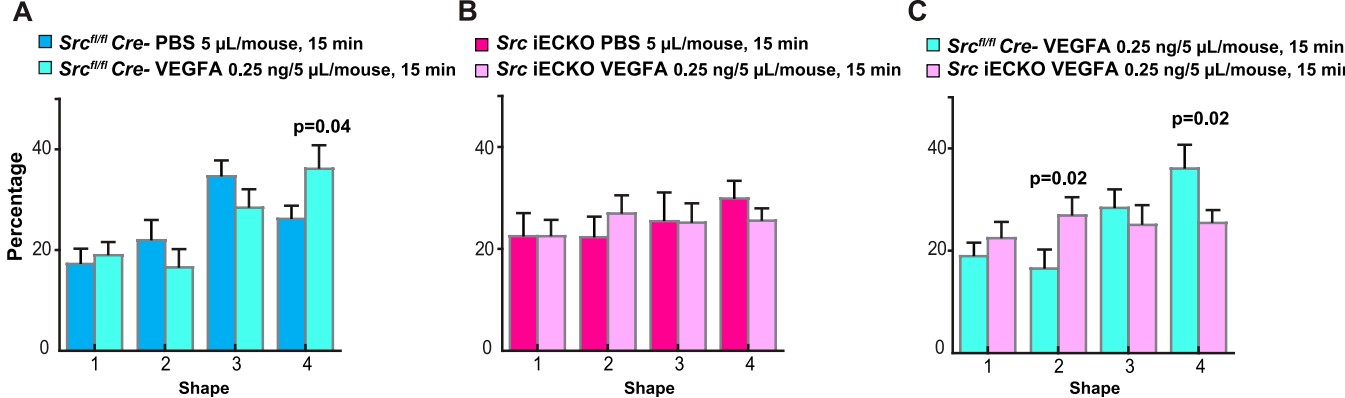

**Figure 6. Role of c-Src in VE-cadherin junctional morphology.**
**(A)** Shape analysis of VE-cadherin fragments in ear dermal lymphatic vessels of Cre negative mice injected with PBS or VEGFA (0.25 ng in 5 $\mu$l). Mice were injected with tamoxifen (20 mg/ml, 100 $\mu$l) for five consecutive days and allowed rest for 2 d before experiment. n = 7 mice/group. $P$ = 0.04 for category 4, $t$ test. **(B)** Shape analysis of VE-cadherin fragments in ear dermal lymphatic vessels of *Src* iECKO (Src$^{fl/fl}$, Cdh5CreER$^{T2}$+) mice injected with PBS or VEGFA (0.25 ng, 5 $\mu$l). Mice were injected with tamoxifen (20 mg/ml, 100 $\mu$l) for five consecutive days and allowed rest for 2 d before experiment. n = 7 mice/group. The $t$ test was used for statistical analysis. **(C)** Comparison of the effect of VEGFA administration between WT (Src$^{fl/fl}$ Cre−) and *Src* iECKO mice injected with VEGFA. n = 7 mice/group $P$ = 0.02 for category 2, $P$ = 0.02 for category 4, $t$ test.

by the unique features of button and zipper-like junctions. Button junctions in initial lymphatic vessels facilitate entry of both fluid and dendritic cells into the lymphatic system (Baluk et al, 2007). Zippering of initial lymphatic junctions in lacteals correlates with enhanced barrier properties, manifested as reduced uptake of chylomicrons (Zhang et al, 2018). However the VE-cadherin dynamics in lymphatic vessels with different junctional morphologies or in the transition between button and zipper junctions has not been studied. VE-cadherin is implicated in regulation of lymphatic vessel permeability in cancer (Dieterich et al, 2022), but how VE-cadherin is affected by the tumor environment, especially in human cancer, remains to be understood. In human colorectal cancer, expression of VEGFC/VEGFR3 is increased, whereas levels of VE-cadherin are lower than in normal tissues (Tacconi et al, 2015). The reduced VE-cadherin expression could be a consequence of transcriptional changes or could indicate an increase in VE-cadherin internalization and degradation akin to the increased VE-cadherin fragmentation, noted here in tumor-challenged mouse models. Better understanding of the lymphatic barrier properties and its regulation, conferred through VE-cadherin and other junctional molecules (Duong & Vestweber, 2020), may provide opportunities for therapeutic applications of tools to stabilize the lymphatic barrier.

# Materials and Methods

## Mice

The *Vegfr2*$^{Y949F/Y949F}$ strain (Li et al, 2016) on the /C57BL/6J background was created using VelociGene technology (Valenzuela et al, 2003) (Regeneron Pharmaceuticals) and maintained by crossing heterozygotes. *Cdh5-CreERT2* mice were provided by Ralf Adams (Max-Planck Institute, Münster, Germany) (Kogata et al, 2006; Wang et al, 2010). c-Src-floxed mice were purchased from the NiceMice National Resource Center for Mutant Mice, Model Animal Research

Center, China, and crossed with the *Cdh5-CreERT2* mice. Cre activity and gene deletion were induced by intraperitoneal injections to mice with 2 mg tamoxifen (Sigma-Aldrich) for 5 d, and mice were used for experiment on day 7 when the deletion efficiency was about 80% (Jin et al, 2022). All animal experiments were repeated at least three independent times with age-matched WT and mutant mice. Cohorts were chosen to ensure reproducibility and to allow stringent statistical analysis. For the different studies, embryos, pups, and young adults (4–8 wk) were used.

## Ethics statement

Animal experiments were carried out in accordance with the ethical permit approved by the Committee on the Ethics of Animal Experiments of the University of Uppsala (permit no. C119/13) and conform to the guidelines from Directive 2010/63/EU of the European Parliament. Anesthesia of the mice in experiments was achieved by inhalation of 2–3% isoflurane. Anesthetic depth was assessed by toe pinch before the procedures. Mice were closely monitored during and after anesthesia. For tissue collection, euthanasia was performed by cervical dislocation.

## Antibodies and growth factors

All antibodies used in this study, their origin, and working concentrations are listed in Table 1. Recombinant human VEGFA$_{165}$ (PeproTech) and human VEGFC (Sigma-Aldrich) were used for in vivo intradermal injections and in vitro experiments.

## Intradermal B16F10 primary tumor and metastasis model

Lentivirus-transduced DsRed-expressing B16F10 cells (kindly provided by Professor David D Schlaepfer, Department of Pathology, La Jolla, CA) were cultured in DMEM GlutaMAX medium (Gibco) containing 10% FCS (Sigma-Aldrich), washed, and resuspended in Matrigel (Becton Dickinson; BD) for inoculation. Mice were

**Table 1.  Antibody information.**

| Antibody name | Manufacturer | Cat. No. | RRID | Dilution |
|---|---|---|---|---|
| Goat anti-mouse CD45 | R&D Systems | AF114 | AB_442146 | 1:200 (IF) |
| Rat anti-mouse CD169 | Bio-Rad | MCA947G | AB_322322 | 1:50 (IF) |
| Rabbit anti-mouse LYVE-1 | ReliaTech GmbH | 103-PA50 | AB_2783787 | 1:500 (IF) |
| Rat anti-mouse LYVE-1 | R&D Systems | MAB 2125 | AB_2138528 | 1:500 (IF) |
| Goat anti-mouse Neuropilin-2 | R&D Systems | AF567 | AB_2155253 | 1:200 (IF) |
| Goat anti-mouse VE-cadherin | R&D Systems | AF1002 | AB_2077789 | 1:250 (IF) |
| Rabbit anti-mouse pY685 VE-cadherin | In-house raised against synthetic phosphopeptide | — | | 1:200 (IF) |
| Goat anti-tdTomato | SICGEN Antibodies | AB8181 | AB_2722750 | 1:200 (IF) |
| Goat anti-mouse Podoplanin | R&D Systems | AF3244 | AB_2268062 | 1:1,000 (WB) |
| Rabbit anti-VEGFR2 | Cell Signaling | 9698 | AB_11178792 | 1:1,000 (WB) |
| Goat anti-mouse VEGFR3 | R&D Systems | AF743 | AB_355563 | 1:1,000 (WB) |
| Mouse anti-GAPDH | Merck Millipore | MAB374 | AB_2107445 | 1:2,000 (WB) |
| Alexa Fluor 488, 555, 647-conjugated secondary antibodies | Life Technologies | — | | 1:500 (IF) |
| HRP-conjugated anti-rabbit IgG | Cytiva | NA934 | AB_772206 | 1:10,000 (WB) |
| HRP-conjugated anti-mouse IgG | Cytiva | NA931 | AB_772210 | 1:10,000 (WB) |
| HRP-conjugated anti-goat IgG | DAKO | P0449 | AB_2617143 | 1:4,000 (WB) |
| BV421 rat Anti-Mouse TER-119 | BD Biosciences | 563998 | AB_2738534 | 1:100 (FACS) |
| PerCP-Cyanine5.5 Rat anti-CD45 | Thermo Fisher Scientific | 45-0451-80 | AB_906233 | 1:100 (FACS) |
| PerCP-Cyanine5.5 Rat anti-CD11b | Thermo Fisher Scientific | 45-0112-82 | AB_953558 | 1:100 (FACS) |

RRID, research resource identifier; IF, immunofluorescence; WB, Western blot; FACS, fluorescence-activated cell sorting.

anesthetized with 3% isoflurane (Isoba), and cells injected in the mouse left ear dermis ($0.4 \times 10^5$ in 5 $\mu$l growth factor–depleted Matrigel; BD) using a 30G insulin syringe (Terumo). Tumor volume was measured with a caliper every other day. At day 7 (D7) after inoculation, ears were collected and used for wholemount immunostaining. At D12, the cervical nodes were collected, weighed, and used for immunostaining or quantitative PCR.

**EO771-CCR7-tdTomato mammary tumor model analysis**

EO771-CCR7-tdTomato cells expressing the C-C chemokine receptor type 7 (CCR7) and the fluorescent protein tdTomato were generated by retroviral transduction. Cells were cultured in RPMI-1640 Hepes (Gibco) supplemented with 10% FBS, 1% L-glutamine, and 1% penicillin-streptomycin (all Gibco) at 37°C, 5% $CO_2$. Tumors were inoculated by injection of $1 \times 10^5$ EO771-CCR7-tdTomato cells in 5 $\mu$l PBS (Gibco) into the fourth mammary fat pad of female mice. Mice were euthanized, and tumor-draining lymph nodes were harvested when tumors reached 10–12 mm in one dimension (size restriction based on ethical permit). Single-cell suspensions of tumor-draining lymph nodes were obtained after digestion in RPMI supplemented with 0.2 mg/ml collagenase P (Sigma-Aldrich), 0.8 mg/ml dispase II (Sigma-Aldrich), and 0.05 mg/ml DNase (Sigma-Aldrich) and incubated with the anti-CD16/CD32 (Invitrogen) antibody to block unspecific binding. Surface antigens were stained with the following specific antibodies: BV421 anti-Ter119 (563998, cloneTER-119; BD Biosciences), PerCp-Cy5.5 anti-CD45 (45-0451-80, clone 30-F11; Thermo Fisher Scientific), PerCp-Cy5.5 anti-CD11b (45-0112-82, clone M1/70; Thermo Fisher Scientific). Dead cells were excluded by SYTOX Blue dead cell staining (Invitrogen). CountBright Absolute Counting Beads (Life Technologies) were added to each sample to calculate total cell numbers. Data were acquired on a BD FACSAriaIII flow cytometer (BD Biosciences) and analyzed with FlowJo software (version 10.6.1; FlowJo, LLC).

**Intravital imaging of lymphatic clearance**

Lymphatic vessel drainage function was assessed by measuring the clearance over time after intradermal skin injection of the infrared probe P20D800 (Karaman et al, 2015). Mice were anesthetized with isoflurane (2%), and 3 $\mu$l of 3 $\mu$M P20D800 was injected intradermally in the ears with a 30G insulin syringe (Terumo). For mice with D7 tumors, injection was performed in the peritumoral area. The mice were then positioned in a whole-animal fluorescence imaging system (NightOWL II; Berthold Technologies), and images were acquired with the following imaging settings: $\lambda_{ex}$: 745 nm, $\lambda_{em}$: 800 nm, and an exposure time of 4 s. Subsequent images were acquired of the ears at 1, 2, 3, 4, 6, and 24 h after injection. Mice were allowed to wake up and move freely between imaging time points. Fluorescence signal intensities were adjusted to baseline ear

signals before injection of tracers to calculate tissue enhancement values. The fluorescence intensity values over time were fit to a one-phase exponential decay model in GraphPad Prism 7.0 software with lymphatic clearance expressed as decay constant k (expressed in h$^{-1}$) or as half-life (expressed in h) using the following equations:

$$Normalized\ Fluorescence\ Intensity = e^{-kt} \qquad (1)$$

$$HalfLife = ln(2)/k \qquad (2)$$

### Quantitative PCR

Cervical nodes from B16F10 tumor-bearing mice at D12 were dissected to remove the fat/connective tissues and weighed. RNA was extracted and purified from the nodes using an RNeasy mini kit (Qiagen). RNA concentrations were measured in a NanoDrop spectrophotometer (Thermo Fisher Scientific) and adjusted to equal concentrations, followed by reverse transcription using SuperScript III (Thermo Fisher Scientific). Quantitative real-time PCR (qRT-PCR) was performed on a Bio-Rad CFX96 real-time PCR machine using SsoAdvanced SYBR Green Supermix (Bio-Rad). The housekeeping gene *Gapdh* (Mm99999915_g1; Thermo Fisher Scientific) was used as an internal control. The comparative Ct method was used to calculate fold difference in gene expression. For further analysis and data visualization, basal gene expression in samples was expressed as the fold-change in gene expression as compared with the unaffected inguinal node. The primer sequences used for *Tyrp1* were as follows:

Forward 5′ GCC CAG CAT CCT TCT TCT CCT CCTG 3′
Reverse 5′ GGT CCC TCA GGT GTT CCA TCG CATA 3′

### In vivo VEGFA/C injections

Mice were anesthetized with isoflurane (2%), and 5 $\mu$l of 50 pg/$\mu$l VEGFA or 5 $\mu$l of 5 ng/$\mu$l VEGFC were injected into the left ear dermis using a 30G insulin syringe (Terumo). An equal volume of PBS was injected into the right ear as a control. The injection site was visually marked using a marker pen. Ears were collected after 15 and 30 min, followed by fixation in 4% PFA and whole-mount immunostaining.

### Isolation of LECs

A modified protocol for LEC isolation was followed (Frye et al, 2018). Mouse embryos were collected at E18.5 and placed in ice-cold DMEM. The dorsal skin was dissected and transferred to ice-cold HBSS supplemented with 10 mM Hepes and 5% FBS. The dissected skin was digested in a mixture of DMEM, 20% FBS, 10 mM Hepes (all from Gibco), 2.5 mg/ml collagenase II, 2.5 mg/ml collagenase IV, and 1 mg/ml deoxyribonuclease (all from Worthington) for 30 min at 37°C, pipetting with a wide-bore transfer pipette every 5 min to assist tissue dissociation. Skin cell suspensions were filtered through a 40-$\mu$m disposable cell strainer (BD Falcon), and cells were centrifuged at 300$g$ for 10 min, suspended in cold PBS with 0.5% BSA, 2 mM EDTA, and incubated with mouse CD45 MicroBeads

(MACS; Miltenyi Biotec) for 15 min at 4°C. Labeled cells were magnetically separated (MACS separator), and CD45-negative cells were collected. The cells were then washed once in PBS with 0.5% BSA, 2 mM EDTA, and first incubated with a primary rabbit anti-mouse LYVE-1 antibody (#11-034; AngioBio) for 40 min at 4°C. Subsequently, the cells were labeled with Anti-rabbit IgG MicroBeads (MACS; Miltenyi Biotec) for 15 min at 4°C and magnetically separated (MACS separator). The isolated LYVE-1–positive cells were lysed in RIPA buffer, and proteins were separated by SDS–PAGE (4–12% gradient gel) (Thermo Fisher Scientific), transferred to nitrocellulose membranes (GE Healthcare), and incubated sequentially with primary and HRP-conjugated secondary antibodies (Table 1). Signals were detected using enhanced chemiluminescence (Cytiva), and images retrieved using Bio-Rad ChemiDocMP and were analyzed using Image Lab software.

### Immunostaining

For adult tissues, ears were split in half, and the inner part of the ear (without cartilage) was used for analyses. For ears with tumors at D7, after splitting the ears and clearing cartilage, the tumor was removed, and ears were fixed in 4% PFA and permeabilized with 0.3% Triton-X/100 for 10 min. After incubation overnight at 4°C in 5% nonfat dry milk/0.3% Triton-X/100 in PBS, tissues were incubated with primary antibodies overnight at 4°C and with secondary antibodies for 2 h at RT before mounting on glass slides. For pY685 VE-cadherin staining, ears were first fixed in 2% PFA for 1 h at RT before the staining procedure. E14.5 embryo back skins were dissected and fixed in 4% PFA. Immunostaining was proceeded as for the adult skin. For lymph node staining, tissues were fixed in 4% PFA overnight at 4°C and dehydrated subsequently in 30% and 15% sucrose at 4°C. Tissues were sectioned in 5-$\mu$m sections and immunostained.

### Imaging, image analysis

Microscopy was done with a Leica TCS SP8 Confocal Microscope with PMT-HyD detectors and LAS X Navigator software version 3.5.2.18963. Images for vessel density and diameter were acquired using a 10x dry/NA 0.3 objective, whereas all other images were obtained using a 63x oil/NA1.3 objective with a further zoom of 2x for capturing metastatic cells and junctions. At least 5–7 fields of view (FOV) per sample were obtained, and all image analyses were performed using ImageJ2 software from NIH. AngioTool (Zudaire et al, 2011) was used to calculate vessel density, which was normalized to the vessel area/FOV, whereas vessel diameter was manually measured using the line tool (ImageJ). Protein intensity measurements were normalized to area of measurement.

### Shape analysis of VE-cadherin fragments

For VE-cadherin junctional quantification, the images were cropped into 20 × 20-$\mu$m blocks, and each block was subjected to segmentation. Objects identified post segmentation were measured for values of circularity and the aspect ratio. The data were exported into MATLAB for further analysis. Using a binning algorithm, in which the boundaries of the bins are defined by the circularity and aspect ratio values, the objects were classified into the four

categories, and the percentage of each junctional class was calculated on the total number of junctions per FOV. Significance was calculated using the one-tailed Wilcoxon rank-sum test.

## Statistical analysis

A $t$ test or Mann–Whitney test was performed to compare means between two groups. Two-way ANOVA was used to compare tumor growth between WT and mutant mice. Two-way ANOVAs with the Tukey's post hoc test was performed when two factors were involved, for example, treatment and junction type. For nonparametric comparison, the Wilcoxon sign-rank test was used. GraphPad Prism 10.1.0 was used for statistical analyses. All quantifications are represented as mean ± SEM. The threshold for statistical significance was set at 0.05.

# Data Availability

Data supporting the findings in this study are included in the main article and associated files. Source data are provided with this study.

# Supplementary Information

# Acknowledgements

We gratefully acknowledge the expert assistance of Marie Hedlund and Joakim Lehrstrand and the support and access to infrastructure at the BioVis facility, Uppsala University. We are grateful for comments and advice from Donald M McDonald, UCSF and Anne Eichmann, Yale School of Medicine. The study was supported by fellowship grants from the Marfan Foundation (Victor A. McKusick Fellowship) and from Uppsala University (Gustav Adolf Johansson and P.O. Zetterling Fellowships). MH Ulvmar was supported by the Swedish Research Council (2016-02492), the Swedish Cancer Foundation (2017/759 and 20 0970 PjF), and the Kjell and Märta Beijer Foundation. This study was further supported by grants to L Claesson-Welsh from the Swedish Research Council (2016-02492 and 2022-00896), the Swedish Cancer Foundation (22 2029 Pj 01 H), the Knut and Alice Wallenberg Foundation (KAW) project grant KAW 2020.0057, The Swedish Foundation for International Cooperation in Research and Higher Education (STINT) (CH2018-7817), and the Fondation Leducq Transatlantic Network of Excellence Grant in Neurovascular Disease (17 CVD 03).

## Author Contributions

M Sáinz-Jaspeado: conceptualization, investigation, methodology, and writing—original draft, review, and editing.
S Ring: investigation and writing—review and editing.
ST Proulx: investigation and writing—review and editing.
M Richards: investigation and writing—review and editing.
P Martinsson: investigation and writing—review and editing.
X Li: investigation and writing—review and editing.
L Claesson-Welsh: conceptualization, supervision, funding acquisition, and writing—original draft, review, and editing.
MH Ulvmar: funding acquisition, investigation, and writing—review and editing.
Y Jin: conceptualization, supervision, investigation, methodology, and writing—original draft, review, and editing.

## Conflict of Interest Statement

The authors declare that they have no conflict of interest.

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
