## [Reviewer comments · Life Science Alliance]

Life Science Alliance

VE-cadherin junction dynamics in initial lymphatic vessels promotes lymph node metastasis

Miguel Sáinz-Jaspeado, Sarah Ring, Steven Proulx, Mark Richards, Pernilla Martinsson, Xiujuan Li, Lena Claesson-Welsh, Maria Ulvmar, and Yi Jin

DOI: <https://doi.org/10.26508/lsa.202302168>

Corresponding author(s): Yi Jin, Uppsala University

Review Timeline:

Submission Date:	2023-05-16
Editorial Decision:	2023-07-11
Revision Received:	2023-11-13
Editorial Decision:	2023-12-07
Revision Received:	2023-12-14
Accepted:	2023-12-15

Transaction Report:

July 11, 2023

Re: Life Science Alliance manuscript #LSA-2023-02168-T

Dr. Yi Jin
Uppsala University
Dept. of Immunology, Genetics and Pathology
Rudbeck Laboratory
Dag Hammarskjöldsv. 20
Uppsala 751 85
Sweden

Dear Dr. Jin,

Thank you for submitting your manuscript entitled "Fragmentation of capillary lymphatic junction facilitates tumor metastasis to lymph nodes" to Life Science Alliance. The manuscript was assessed by expert reviewers, whose comments are appended to this letter. We invite you to submit a revised manuscript addressing the Reviewer comments.

Thank you for this interesting contribution to Life Science Alliance. We are looking forward to receiving your revised manuscript.

Sincerely,

Eric Sawey, PhD
Executive Editor
Life Science Alliance
<http://www.lsa-journal.org>

B. MANUSCRIPT ORGANIZATION AND FORMATTING:

Reviewer #1 (Comments to the Authors (Required)):

1.
Sáinz-Jaspeado et al. have submitted the manuscript "Fragmentation of capillary lymphatic junction facilitates tumor metastasis to lymph nodes". The manuscript describes the role of VEGFA-pVEGFR2 signalling in lymphatic vessels during cancer progression and lymph node metastasis, and focuses exactly on VEGFR2 Tyr949 phosphorylation. They discovered fragmentation of LEC junctions in lymphatic capillaries adjacent to melanoma tumors, which was not seen in the VEGFR2-Y949F phosphorylation deficient knock-in (KI) mouse model. The study is a continuum to their previous work related to the VEGFR2-Y949 in blood vessels, and here they deepen the understanding of VEGFR2 phosphorylation in the regulation of adherens junction stability specifically in the lymphatic vessels.

2.
Figure 1 compares dermal lymphatic capillaries in E14.5 mouse embryos and in adult wt and KI mice, VEGFR levels at P 10 as well as lymphatic drainage in adult wt and KI mice. No major differences are observed, except for the increased lymphatic vessel density in the KI mice as compared to wt. Thus, the dermal lymphatics appear develop largely normally in the KI mice.

Figure 2 focuses on lymph node (LN) metastasis in the wt vs KI mice using two models. It is shown that while tumor growth is similar in both wt and KI mice, the metastases are less in KI mice as compared to wt. In Fig. 2H, the authors should include an image of a LN also from the KI mice. In addition, could the results be explained by decreased tumor lymphangiogenesis in wt vs KI mice? Lower tumor interstitial pressure in KI mice, due to reduced leakage, might result in decreased lymphangiogenesis in the periphery. This could be confirmed using immunohistochemical staining and subsequent quantification of the lymphatic markers in tumors and surrounding tissue.

Figure 3 shows that melanoma-induced VE-cadherin loss in peritumoral lymphatic capillaries is decreased in KI mice. The results would be strengthened by analysis of VE-cadherin protein / mRNA levels in tissues from wt vs KI mice +/- cancer.

Figure 4 shows that VEGF impairs VE-cadherin positive junctions in the lymphatic capillaries from wild type but not KI mice. Since VEGF-induced loss of VE-cadherin is presented as a mechanism in the tumor models, VEGF levels should be analyzed in wt vs KI mice +/- cancer.

Figure 5 studied the contribution of SFK on VEGF-induced Ve-cadherin loss in isolated LECs from wt and KI mice. By utilizing a conditional SFK knock-out mice they showed that VEGF-induced VE-cadherin impairment was mediated by SFK.

3.
See points made above for Fig 2, 3 and 4.

Reviewer #2 (Comments to the Authors (Required)):

Summary: The goal of this study was to test the hypothesis that tumor metastasis can be facilitated by fragmentation of junctions in lymphatic vessels, akin to the phenomenon reported for blood vasculature. The authors tested the hypothesis in melanoma and breast cancer models in a model with functionally defective VEGFR-2 that lacks the ability to phosphorylate junctional proteins (Y949F) and compared the data to wild-type (WT). They found statistically significant higher presence of tumor cells in lymph node (LN) in WT compared with Y949F which correlated with reduced lymph clearance representing lymph flow. The latter is presumably due to better junctional stability in Y949F animals which prevents stasis and facilitates moving fluid forward (this is my understanding, the underlying mechanism is not exactly spelled out). Characterization of lymphatic endothelial cell junctions in vivo and in vitro showed that either tumors or their derived VEGF-A or VEGF-C increase fragmentation of junctions to short segments which may increase intravasation of tumor cells. This event was also shown to be c-Src dependent, as it is in blood vessels.

Major critiques:

- 1) While the study describes a potentially important mechanistic aspect of lymphatic metastasis, some data are not very convincing and some - are not well presented. Specifically, the differences between WT and VEGFR-2 mutant mice seem to be miniscule and hard to interpret in terms of the exact metastatic burden. This is because qPCR method is based on ratio and normalization, but in this case, it is unclear how Ct values were normalized among different animals, what was taken as fold-change of 1.0; and how differences in the number, protein contents, and sizes of different lymph nodes were accounted for. This method does not allow to translate Ct values (assuming they are properly normalized per housekeeping genes) into tumor cell equivalents, and with such a small difference shown in Fig. 2D, it is hard to be convinced that that tumors in WT mice metastasize more efficiently than in Y949F animals. Also, the difference in several mgs in LN weight, although appears to be statistically significant, it is difficult to be taken as representative of all tumors due to an unusual site of tumor implantation, a very early tumor stage, and only minor changes despite using a highly sensitive qPCR technique. Indeed, the breast model did not show such changes.
- 2) E0771 model produced more convincing results but the problem here (as with all other models) that analyses were performed only in one time point, so the difference in ~100 cells per LN (based on Fig. 2G) might be not sustainable at later points. In other words, even if the differences in invasion of lymphatics have some impact at early tumor stages, this might not have a significant effect on overall metastatic burden and spread to LNs beyond the sentinel nodes and other normal organs. The kinetics analysis of one or both models would be more convincing than an isolated time point.
- 3) The main claim that reduced clearance directly corresponds to increased vascular invasion should be confirmed by additional means. It is logical to suggest that impairment of vascular integrity through junctional fragmentation might result in both phenomena, but at the same time, other known factors can produce opposing effects. For instance, reduced flow can decrease the number of LN-bound migrating tumor cells or reduce number of exiting immune stimulatory cells thus promoting tumor cell kill. The presented data obtained at one early time point and showing rather minor differences, although supportive of the claim, are not sufficient to draw a bold conclusion (page 4, the second paragraph from bottom, claim about "suppression" of metastasis).
- 4) Data presentation for "Fragmentation category" is very confusing. The data should be presented as bars for each category rather than linking all four categories with a line. There is no reason to connect the percentage of "long" or "very long" fragments to "short" or "very short". Once again, an additional method confirming junctional fragmentation in tumor-associated lymphatic vessels mediated by tumor-secreted factors would significantly strengthen the study conclusions.
- 5) The Discussion section contains very lengthy discussions on morphology of lymphatic vessels in different normal organs but no reference to either tumors or metastasis. How important is the described fragmentation of junctions in lymphatic vessels to the metastatic process? How significant it is for human cancers given other factors that can influence intravasation, transmigration of tumor cells through lymphatic barrier at non-junctional sites and other parameters. Does the efficiency of early invasion determine the overall outcome? Does fragmentation occur in non-metastatic tumors? Given the focus of the study, it seems that Discussion should address at least some of these questions while shortening a bit less relevant literature analyses of lymphatic morphology outside of the tumor context.

Minor critiques:

- 1) VEGF-A, VEGF-C, VEGFR-3 and VEGFR-2 should be spelled out with dash to denote proteins.
- 2) TRP1 should be identified as tyrosinase related protein-1 and its proper gene name TYRP1.
- 3) Fig. 4A and Fig. 4G look identical and it takes some time to understand that one depicts 15 min assessment whereas the other shows results for 30 min. Please indicate the time difference in the graphs in addition to description in figure legends.
- 4) Classification of analyzed fragments and description of each category should be described in a more quantifiable manner (e.g., what is the numerical difference between each of 1-4 categories).
- 5) The list of antibodies in Supplementary Table 1 does not accurately correspond to used antibodies in the study. It is not clear whether Tomato was detected by red fluorescence or by anti-Tomato antibody. It is not clear what was used for detection of CD169 in LNs. None of the presented figures shows staining with anti-neuropilin-2 IgG listed in the Table. It is not clear what was the use of HRP-conjugated secondary antibodies (all images are from IF).
- 6) Please confirm that E0771 tumor cells were injected without Matrigel. The standard method of tumor implantation does require Matrigel.
- 7) Results state that B16 cells were injected intradermally but the Methods describe it as a subcutaneous model. Please reconcile.

Re: Rebuttal letter Life Science Alliance manuscript #LSA-2023-02168-T

Dear Editor,

We appreciate the reviewers' constructive and insightful criticisms. An important addition to the study, inspired by these comments, is shown in the new Figures 4 and 5, where VE-cadherin fragment lengths have been measured specifically at junctions. The data clearly demonstrates "zippering" both of tumor-proximal lymphatics and dermal initial lymphatics after VEGFA-injection. Zippering occurred in both genotypes studied, wildtype (WT) and *Vegfr2*^{Y949F/Y949F} mice. In contrast, the automated quantification of VE-cadherin shapes (based on aspect ratio/circularity) shown in the original submission, was done throughout the cell. The VE-cadherin shape change occurred only in the WT and not in the *Vegfr2*^{Y949F/Y949F} mutant when analyzing tumor-proximal lymphatics and initial lymphatics in the VEGFA-injected mouse ear. We speculate that the shape change, i.e. VE-cadherin internalization/fragmentation provides a potential mechanism for the difference in tumor cell extravasation and metastatic spread between the WT and the *Vegfr2*^{Y949F/Y949F} mice, as outlined in the new Discussion.

In summary, the figures now show the following:

- Figures 1 and 2 remain essentially as before.
- Figure 3 shows the clearance of an interstitial tracer and tumor cell intravasation into lymphatics, in the two genotypes.
- Figures 4 and 5 combine the new measurements of VE-cadherin dynamics at the junctions, with the previously shown shape change throughout the cells in tumor-proximal vs tumor-distal lymphatics (Figure 4) and after VEGFA-injection (Figure 5).
- Figure 6 shows that the Src pathway is required for VEGFA-induced VE-cadherin shape change
- Supplemental Figure 1 which is new, shows lymphatic vessel density and expression levels of *Vegfa* and *Cdh5*.
- Supplemental Figure 2 shows VE-cadherin phosphorylation in lymphatics.
- Supplementary Figure 3 shows VE-cadherin shape change in response to VEGFC.
- The previous Supplementary Figure 2 which showed inflammation when a very high dose of VEGFA was administered has been removed as it appeared redundant in the revised version.

The text has been adjusted throughout.

Please see below for our point-by-point responses.

Reviewer #1

1. Sáinz-Jaspeado et al. have submitted the manuscript "Fragmentation of capillary lymphatic junction facilitates tumor metastasis to lymph nodes". The manuscript describes the role of VEGFA-pVEGFR2 signalling in lymphatic vessels during cancer progression and lymph node metastasis, and focuses exactly on VEGFR2 Tyr949 phosphorylation. They discovered fragmentation of LEC junctions in lymphatic capillaries adjacent to melanoma tumors, which was not seen in the VEGFR2-Y949F phosphorylation deficient knock-in (KI) mouse model. The study is a continuum to their previous work related to the VEGFR2-Y949 in blood vessels, and

here they deepen the understanding of VEGFR2 phosphorylation in the regulation of adherens junction stability specifically in the lymphatic vessels.

Response: We would like to thank the reviewer for the comments and questions which have led to generation of new data and improvement of this study.

2. Figure 1 compares dermal lymphatic capillaries in E14.5 mouse embryos and in adult wt and KI mice, VEGFR levels at P 10 as well as lymphatic drainage in adult wt and KI mice. No major differences are observed, except for the increased lymphatic vessel density in the KI mice as compared to wt. Thus, the dermal lymphatics appear develop largely normally in the KI mice.

Figure 2 focuses on lymph node (LN) metastasis in the wt vs KI mice using two models. It is shown that while tumor growth is similar in both wt and KI mice, the metastases are less in KI mice as compared to wt.

In Fig. 2H, the authors should include an image of a LN also from the KI mice.

Response: The lymph node picture from a WT mouse in Fig. 2H is to validate lymphatic metastasis in this tumor model. We could not provide a LN image from KI mice because, unfortunately, all lymph nodes collected in this study have been used for quantitative analysis of tumor cell counts by using FACS (Fig. 2G). To again isolate lymph nodes from EO771 challenged female mice would have taken more time than allowed due to current problems in the facility with poor birth and small/lost litters and therefore inability to expand the *Vegfr2*^{Y949F/Y949F} colony.

In addition, could the results be explained by decreased tumor lymphangiogenesis in wt vs KI mice? Lower tumor interstitial pressure in KI mice, due to reduced leakage, might result in decreased lymphangiogenesis in the periphery. This could be confirmed using immunohistochemical staining and subsequent quantification of the lymphatic markers in tumors and surrounding tissue.

Response: We did LYVE1 immunostaining in the ears from B16F10 tumor engrafted mice and quantified the lymphatic vessel density at the tumor periphery. No significant difference was shown between WT and *Vegfr2*^{Y949F/Y949F} mice. The data is now shown in the Supplementary Figure 1A,B.

Figure 3 shows that melanoma-induced VE-cadherin loss in peritumoral lymphatic capillaries is decreased in KI mice. The results would be strengthened by analysis of VE-cadherin protein / mRNA levels in tissues from wt vs KI mice +/- cancer.

Response: We have now analyzed VE-cadherin patterns in the tumor proximal and tumor distal (healthy) region, see Fig. 4A-D and managed to obtain higher resolution and more representative images. The results show zippering of lymphatic VE-cadherin junctions in the tumor proximal lymphatics. The fluorescent intensity of VE-cadherin is not different between WT and KI mice. In addition we analyzed *Cdh5* transcript levels in B16F10 melanoma tumors from WT and *Vegfr2*^{Y949F/Y949F} mice. There is a trend of decreased *Cdh5* expression is shown in KI mice, however the difference is not statistically significant (Supplementary fig. 1D).

Figure 4 shows that VEGF impairs VE-cadherin positive junctions in the lymphatic capillaries

from wild type but not KI mice. Since VEGF-induced loss of VE-cadherin is presented as a mechanism in the tumor models, VEGF levels should be analyzed in wt vs KI mice +/- cancer.

Response: We performed qPCR to determine the expression of *Vegfa* in B16F10 tumors from WT and *Vegfr2*^{Y949F/Y949F} mice and no significant difference was detected (Supplementary fig. 1C).

Figure 5 studied the contribution of SFK on VEGF-induced VE-cadherin loss in isolated LECs from wt and KI mice. By utilizing a conditional SFK knock-out mice they showed that VEGF-induced VE-cadherin impairment was mediated by SFK.

3. See points made above for Fig 2, 3 and 4.

Reviewer #2 (Comments to the Authors (Required)):

Summary: The goal of this study was to test the hypothesis that tumor metastasis can be facilitated by fragmentation of junctions in lymphatic vessels, akin to the phenomenon reported for blood vasculature. The authors tested the hypothesis in melanoma and breast cancer models in a model with functionally defective VEGFR-2 that lacks the ability to phosphorylate junctional proteins (Y949F) and compared the data to wild-type (WT). They found statistically significant higher presence of tumor cells in lymph node (LN) in WT compared with Y949F which correlated with reduced lymph clearance representing lymph flow. The latter is presumably due to better junctional stability in Y949F animals which prevents stasis and facilitates moving fluid forward (this is my understanding, the underlying mechanism is not exactly spelled out). Characterization of lymphatic endothelial cell junctions in vivo and in vitro showed that either tumors or their derived VEGF-A or VEGF-C increase fragmentation of junctions to short segments which may increase intravasation of tumor cells. This event was also shown to be c-Src dependent, as it is in blood vessels.

Response: We would like to thank the reviewer for the comments and questions which have led to generation of new data and improvement of this study. We have now tried our best to increase the clarity of the presentation.

Major critiques:

1) While the study describes a potentially important mechanistic aspect of lymphatic metastasis, some data are not very convincing and some - are not well presented. Specifically, the differences between WT and VEGFR-2 mutant mice seem to be miniscule and hard to interpret in terms of the exact metastatic burden. This is because qPCR method is based on ratio and normalization, but in this case, it is unclear how Ct values were normalized among different animals, what was taken as fold-change of 1.0; and how differences in the number, protein contents, and sizes of different lymph nodes were accounted for. This method does not allow to translate Ct values (assuming they are properly normalized per housekeeping genes) into tumor cell equivalents, and with such a small difference shown in Fig. 2D, it is hard to be convinced that that tumors in WT mice metastasize more efficiently than in Y949F animals. Also, the difference in several mgs in LN weight, although appears to be statistically significant, it is difficult to be taken as

representative of all tumors due to an unusual site of tumor implantation, a very early tumor stage, and only minor changes despite using a highly sensitive qPCR technique. Indeed, the breast model did not show such changes.

Response: We apologize for the poor presentation of the qPCR data in Fig. 2D in the original manuscript. We reanalyzed the qPCR data and replaced it with a new graph. The data was analyzed by the ddCt method using mouse housekeeping gene *Rpl19* as control. mRNA levels of *Tyrp1* in the cervical lymph nodes from B16F10-challenged WT and *Vegfr2*^{Y949F/Y949F} mice were normalized to the *Tyrp1* level in unaffected inguinal lymph nodes collected from WT mice.

The ear dermis was chosen as the site of injection because it restricts local growth of tumor volume and promotes spread to sentinel lymph nodes. As we are sure the reviewer agrees, there are not many transplantable, C57Bl/6 compatible metastatic models to choose from that also are amenable to imaging. We could have tried to place the B16F10 tumors on the flank, surgically remove the tumors when at 1cm³ (ethical restrictions do not allow bigger tumors) and wait for lung metastasis to become established. We have attempted this strategy but were not sufficiently proficient at resecting the primary tumors in a consistent manner, giving rise to quite uneven regrowth. We know this is also a considerable problem in other laboratories. We agree that the extent of lymphatic metastasis is quite variable between individuals as shown in the new Fig. 2D and the images of lymph nodes presented below (Rebuttal Fig.1). However we respectfully disagree with attributing the difference between WT and *Vegfr2*^{Y949F/Y949F} mice in metastatic dissemination to lymph nodes, to random statistically variation in the two tumor models, the different methods shown in Fig. 2. Taken together we are confident that the metastasis in lymph nodes is reduced in the *Vegfr2*^{Y949F/Y949F} mutant mice, especially for the incidence of large lymphatic metastasis.

Rebuttal Figure 1. Images of cervical lymph nodes collected at day 12 after implantation of B16F10 melanoma cells in the ear dermis of WT and *Vegfr2*^{Y949F/Y949F} mice. Note the metastasis of the melanoma cells that are visible in some of the lymph nodes.

2) E0771 model produced more convincing results but the problem here (as with all other models) that analyses were performed only in one time point, so the difference in ~100 cells per LN (based on Fig. 2G) might be not sustainable at later points. In other words, even if the differences in invasion of lymphatics have some impact at early tumor stages, this might not have a significant effect on overall metastatic burden and spread to LNs beyond the sentinel

nodes and other normal organs. The kinetics analysis of one or both models would be more convincing than an isolated time point.

Response: The ~100 cells difference per LN represent a 50% decrease in metastatic burden in *Vegfr2*^{Y949F/Y949F} mice compared to WT at this stage of tumor growth. In a separate study to describe and validate the EO771 model (to be submitted; M H. Ulvmar, senior author), we did a kinetic analysis of lymphatic metastasis at day 12 and day 20 after tumor cell implantation. The result showed that Tomato+ tumor cells could be detected by FACS analysis in WT inguinal lymph nodes as early as day 12 (about 40 cell counts per LN) and the cell count significantly increased at day 20 to a mean of about 200 cell counts per LN (Rebuttal Fig. 2). Therefore the difference of 100 cells in this model does represent a solid reduction in metastastatic spread.

To have this tumor model grow longer is not feasible; it severely increases the risk of wounding at the primary tumor site, and the tumors grow too large. We are permitted to have tumors grow to 0.5 cm³/site for this model, and of course with time, the tumors start to grow exponentially. Therefore for ethical reasons we could not extend much beyond the 20 days of tumor growth shown in this study. Thus, although we in principal agree with the reviewer that it is important to perform kinetic analysis, lymph node spread would not be possible to study at later time points for ethical reasons and at earlier time points, the number of tumor cells/LN is quite small. Day 20 appears to be an optimal time point for analysis.

Rebuttal Figure 2. FACS analysis of Tomato+ EO771-CCR7 tumor cells in inguinal lymph nodes at day 12 (A) and day 20 (B) after implantation of the tumor cells in the mammary fat pad of female mice.

3) The main claim that reduced clearance directly corresponds to increased vascular invasion should be confirmed by additional means. It is logical to suggest that impairment of vascular integrity through junctional fragmentation might result in both phenomena, but at the same time, other known factors can produce opposing effects. For instance, reduced flow can decrease the number of LN-bound migrating tumor cells or reduce number of exiting immune stimulatory cells thus promoting tumor cell kill. The presented data obtained at one early time point and showing rather minor differences, although supportive of the claim, are

not sufficient to draw a bold conclusion (page 4, the second paragraph from bottom, claim about "suppression" of metastasis).

Response: We apologize; it was not our intention to overstate the results or conclude that changes in clearance directly corresponds to changes in metastatic spread. We agree that clearance of interstitial fluid and intravasation of tumor cells into lymphatics can be through different mechanisms. The clearance data and tumor cells intravasation data are now put together in a new Figure 3 to show that the Y949F mutation led to improved lymphatic drainage and reduced transmigration of tumor cells into lymphatic vessels, indicating an enhanced peritumoral lymphatic barrier compared to the WT. We have toned down the claim about suppression of metastasis throughout.

4) Data presentation for "Fragmentation category" is very confusing. The data should be presented as bars for each category rather than linking all four categories with a line. There is no reason to connect the percentage of "long" or "very long" fragments to "short" or "very short". Once again, an additional method confirming junctional fragmentation in tumor-associated lymphatic vessels mediated by tumor-secreted factors would significantly strengthen the study conclusions.

Response: We thank the reviewer for this important request. We have now reanalyzed the data using an additional method. The VE-cadherin analysis in the original manuscript was meant to classify in an automated manner, the shapes of the VE-cadherin fragment based on aspect ratio and circularity. It could not analyze the actual length of the fragments. Therefore, we realize that the description of "long" and "short" fragments with this classification was not appropriate; instead, we now refer to shape categories. A more detailed description of the classification has been added (page 5, second paragraph from the bottom). All the graphs regarding the shape analysis have been changed to bar graphs.

Moreover, in the new junctional analysis, we focused on the number and lengths of VE-cadherin fragments at the endothelial cell junctions of initial lymphatic vessels. This analysis led to the important insight that zippering of lymphatic junctions is established in tumor-proximal lymphatics (Fig. 4A-D) and upon VEGF injection in the ear dermis (Fig. 5A-D). The phenomenon of lymphatic junction zippering by VEGFA signaling has been well studied recently (Zarkada, Chen et al., 2023, Zhang, Zarkada et al., 2018) and we now report on zippering also in the peritumoral lymphatics, which as far as we are aware has not been previously observed.

Junctional zippering was accompanied by an increase in pan-cellular circular/low aspect ratio VE-cadherin shapes in the wild-type mouse lymphendothelium, indicating increased VE-cadherin dynamics, but not in mice expressing the Y949F mutant VEGFR2. In contrast, we show that the Y949 site in VEGFR2 is dispensable for VEGFA-induced zippering of lymphatic junctions in agreement with data in Zarkada et al., 2023. However, importantly, signaling downstream of Y949 in VEGFR2 is required for VEGFA/VEGFC-induced internalization and turnover of VE-cadherin.

5) The Discussion section contains very lengthy discussions on morphology of lymphatic vessels in different normal organs but no reference to either tumors or metastasis. How

important is the described fragmentation of junctions in lymphatic vessels to the metastatic process? How significant it is for human cancers given other factors that can influence intravasation, transmigration of tumor cells through lymphatic barrier at non-junctional sites and other parameters. Does the efficiency of early invasion determine the overall outcome? Does fragmentation occur in non-metastatic tumors? Given the focus of the study, it seems that Discussion should address at least some of these questions while shortening a bit less relevant literature analyses of lymphatic morphology outside of the tumor context.

Response: We have amended the Discussion. We would like to point out that the properties of tumor lymphatics was also described in the Introduction.

Minor critiques:

1) VEGF-A, VEGF-C, VEGFR-3 and VEGFR-2 should be spelled out with dash to denote proteins.

Response: We have adopted the conventional style of showing gene names in italics (*Cdh5*, *Vegfa*, *Vegfr2*, *Tyrp1* etc for mouse genes) and proteins in their non-italic capitalized versions. We are not aware that the hyphen is a required designation for proteins but if this is a journal style requirement we will adjust.

2) TRP1 should be identified as tyrosinase related protein-1 and its proper gene name TYRP1.

Response: TRP1 has been changed to *Tyrp1* (rather than TYRP1, to show that we are referring to a mouse gene).

3) Fig. 4A and Fig. 4G look identical and it takes some time to understand that one depicts 15 min assessment whereas the other shows results for 30 min. Please indicate the time difference in the graphs in addition to description in figure legends.

Response: The time point for all the VEGF injection experiments has been added to the graphs.

4) Classification of analyzed fragments and description of each category should be described in a more quantifiable manner (e.g., what is the numerical difference between each of 1-4 categories).

Response: A detailed description has been added in the text as below in page 5, second paragraph from the bottom.

“Four categories of VE-cadherin shapes were defined: category 1; fragments with aspect ratio above the upper quartile (Q3) and circularity between 0-0.25, category 2; aspect ratio Q2-Q3 and circularity 0.25-0.5, category 3; aspect ratio Q1-Q2 and circularity 0.5-0.75, category 4; aspect ratio below Q1 and circularity 0.75-1 (Figure 4E). The small round shapes in categories 3-4 may result from dynamic turnover of VE-cadherin, i.e. internalization (Bentley, Franco et al., 2014).”

5) The list of antibodies in Supplementary Table 1 does not accurately correspond to used antibodies in the study. It is not clear whether Tomato was detected by red fluorescence or by anti-Tomato antibody. It is not clear what was used for detection of CD169 in LNs. None of the presented figures shows staining with anti-neuropilin-2 IgG listed in the Table. It is not clear what was the use of HRP-conjugated secondary antibodies (all images are from IF).

Response: tdTomato in the image of lymph node (Fig. 2H) was detected by antibody staining, although in FACS no antibody was used. Tdtomato antibody information has been added to Supplementary table 1. CD169 antibody information has also been added to Supplementary table 1. Anti-NRP2 was used in Figure 1A. HRP-conjugated secondary antibodies were used in the western blots in Figure 1G.

6) Please confirm that E0771 tumor cells were injected without Matrigel. The standard method of tumor implantation does require Matrigel.

Response: For the B16F10 mdel, we used Matrigel but for the EO771 cell implantation the protocol does not require Matrigel.

7) Results state that B16 cells were injected intradermally but the Methods describe it as a subcutaneous model. Please reconcile.

Response: We apologize. The description in Methods has been corrected; intradermal injection was applied.

References

- Bentley K, Franco CA, Philippides A, Blanco R, Dierkes M, Gebala V, Stanchi F, Jones M, Aspalter IM, Cagna G, Westrom S, Claesson-Welsh L, Vestweber D, Gerhardt H (2014) The role of differential VE-cadherin dynamics in cell rearrangement during angiogenesis. *Nat Cell Biol* 16: 309-21
- Zarkada G, Chen X, Zhou X, Lange M, Zeng L, Lv W, Zhang X, Li Y, Zhou W, Liu K, Chen D, Ricard N, Liao J, Kim YB, Benedito R, Claesson-Welsh L, Alitalo K, Simons M, Ju R, Li X et al. (2023) Chylomicrons Regulate Lacteal Permeability and Intestinal Lipid Absorption. *Circ Res* 133: 333-349
- Zhang F, Zarkada G, Han J, Li J, Dubrac A, Ola R, Genet G, Boye K, Michon P, Kunzel SE, Camporez JP, Singh AK, Fong GH, Simons M, Tso P, Fernandez-Hernando C, Shulman GI, Sessa WC, Eichmann A (2018) Lacteal junction zippering protects against diet-induced obesity. *Science* 361: 599-603

December 7, 2023

RE: Life Science Alliance Manuscript #LSA-2023-02168-TR

Dr. Yi Jin
Uppsala University
Dept. of Immunology, Genetics and Pathology
Rudbeck Laboratory
Dag Hammarskjöldsv. 20
Uppsala 751 85
Sweden

Dear Dr. Jin,

Thank you for submitting your revised manuscript entitled "VE-cadherin junction dynamics in initial lymphatic vessels promotes lymph node metastasis". We would be happy to publish your paper in Life Science Alliance pending final revisions necessary to meet our formatting guidelines.

- please add the Twitter handle of your host institute/organization as well as your own or/and one of the authors in our system
- please update your callouts for the Supplementary Figures in the manuscript: Suppl Fig 2 A, B

A. FINAL FILES:

B. MANUSCRIPT ORGANIZATION AND FORMATTING:

Sincerely,

Reviewer #1 (Comments to the Authors (Required)):

The authors have refined the manuscript and no additional comments are needed.

Reviewer #2 (Comments to the Authors (Required)):

The authors addressed the critiques raised by original version. This reviewer has no additional comments.

December 15, 2023

RE: Life Science Alliance Manuscript #LSA-2023-02168-TRR

Dr. Yi Jin
Uppsala University
Dept. of Immunology, Genetics and Pathology
Rudbeck Laboratory
Dag Hammarskjöldsv. 20
Uppsala 751 85
Sweden

Dear Dr. Jin,

Thank you for submitting your Research Article entitled "VE-cadherin junction dynamics in initial lymphatic vessels promotes lymph node metastasis". It is a pleasure to let you know that your manuscript is now accepted for publication in Life Science Alliance. Congratulations on this interesting work.

DISTRIBUTION OF MATERIALS:

Again, congratulations on a very nice paper. I hope you found the review process to be constructive and are pleased with how the manuscript was handled editorially. We look forward to future exciting submissions from your lab.

Sincerely,
